

# Development of a General Calibration Model and Long-Term Performance Evaluation of Low-Cost Sensors for Air Pollutant Gas Monitoring

Carl Malings[1], Rebecca Tanzer[1], Aliaksei Hauryliuk[1], Sriniwasa P.N. Kumar[1], Naomi Zimmerman[2],
Levent B. Kara[3], Albert A. Presto[1], and R. Subramanian[1]

[1]Center for Atmospheric Particle Studies, Carnegie Mellon University, Pittsburgh, Pennsylvania, 15213, USA
[2]Department of Mechanical Engineering, University of British Columbia, Vancouver, British Columbia, V6T 1Z4, Canada
[3]Department of Mechanical Engineering, Carnegie Mellon University, Pittsburgh, Pennsylvania, 15213, USA

*Correspondence to*: Carl Malings (cmalings@andrew.cmu.edu)

**Abstract.** Assessing the intra-city spatial distribution and temporal variability of air quality can be facilitated by a dense network of monitoring stations. However, the cost of implementing such a network can be prohibitive if traditional high-quality, expensive monitoring systems are used. To this end, the Real-time Affordable Multi-Pollutant (RAMP) monitor has been developed, which can measure up to five gases including the criteria pollutant gases carbon monoxide (CO), nitrogen dioxide ($NO_2$), and ozone ($O_3$), along with temperature and relative humidity. This study compares various algorithms to
calibrate the RAMP measurements including linear and quadratic regression, clustering, neural networks, Gaussian processes, and random forests. Using data collected by more than sixty RAMP monitors over periods ranging up to eighteen months, it was found that quadratic regression models or a hybrid of random forest and linear models tend to be the most effective calibration models overall. In specific cases, other types of models can have comparable or even superior performance. Furthermore, generalized calibration models may be used instead of individual models with only a small reduction in overall
performance. For long-term deployments, it is recommended that new models be developed each year, due to the noticeable change in performance when models for one year were used for processing data collected in the subsequent year. This makes annually-developed generalized calibration models even more useful since only a subset of deployed monitors are needed to build these models. These results will help guide future efforts in the calibration and use of low-cost sensor systems worldwide.

## 1 Introduction

Current regulatory methods for assessing urban air quality rely on a small network of monitoring stations providing highly precise measurements (at a commensurately high setup and operating cost) of specific air pollutants (e.g. Snyder et al., 2013). The United States Environmental Protection Agency (EPA) determines compliance with national air quality standards at the county level using data collected by local monitoring stations. Many rural counties have at most a single monitoring site; urban counties may be more densely instrumented, though not at the neighborhood scale. For instance, the Allegheny County Health

Department (ACHD) maintains a network of ten monitoring stations which collect continuous and/or 24-hour data for the two-



thousand-square-kilometer Allegheny County (with a population of 1.2 million) in Pennsylvania, USA, with only one of these stations providing continuous data for all EPA criteria pollutants listed in the National Ambient Air Quality Standards (NAAQS) (Hacker, 2017). However, air pollutant concentrations can vary greatly even within urban areas due to the large number and variety of sources (Marshall et al., 2008; Karner et al., 2010; Tan et al., 2014). This variability could lead to

inaccurate estimates of air quality based on these sparse monitoring data (Jerrett et al., 2005).

One approach to increasing the spatial resolution of air quality data is the use of dense networks of low-cost sensor packages. Low-cost monitors are instruments which combine one or more comparatively inexpensive sensors (typically electrochemical or metal oxide sensors) with independent power sources and wireless communication systems. This allows larger numbers of monitors to be employed at a similar cost to a more traditional monitoring network as described above. The general goals of

low-cost sensing include supplementing existing regulatory networks, monitoring air quality in areas that have lacked this in the past (for example in developing countries), and increasing community involvement in air quality monitoring through the provision of sensors and the resulting data to community volunteers to support more informed public decision-making and engagement in air quality issues (Snyder et al., 2013; Loh et al., 2017; Turner et al., 2017). Several pilot programs of low-cost sensor network deployment have been attempted, in Cambridge, UK (Mead et al., 2013), Imperial Valley, California (Sadighi

et al., 2017; English et al., 2017), and Pittsburgh, Pennsylvania (Zimmerman et al., 2018).

There are several trade-offs resulting from the use of low-cost sensors. These sensors tend to have lower signal-to-noise ratios than regulatory-grade instruments at typical ambient concentrations due to cross-sensitivities to other pollutants and dependence of the sensor response to ambient temperature and humidity (Popoola et al., 2016). These interactions are often nonlinear, meaning that linear regression models developed under controlled laboratory conditions are often insufficient to

accurately translate the raw sensor responses into concentration measures (Jiao et al., 2016). Due to the variety of interactions and atmospheric conditions which can affect sensor performance, laboratory calibrations are insufficient to cover the range of conditions to which the sensor will be exposed. Field calibrations of the sensors are thus necessary, with the sensors being collocated with highly accurate regulatory-grade instruments. Various calibration methods that have been explored include the determination of sensor calibrations from physical and chemical principles (Masson et al., 2015), higher-dimensional models

to capture nonlinear interactions (Cross et al., 2017), and nonparametric approaches including artificial neural networks (Spinelle et al., 2015) and nearest neighbor clustering (Hagan et al., 2017). Recent work by our group compared lab-based linear calibration models with multiple linear regression and non-parametric random forest algorithms based on ambient collocations (Zimmerman et al. 2018). The machine learning algorithm using random forests on ambient collocation data enabled low-cost electrochemical sensor measurements to meet EPA data quality guidelines for hot spot detection and personal

exposure for $NO_2$ and supplemental monitoring for CO and ozone (Zimmerman et al., 2018).

There remain several unanswered questions with respect to the calibration of data collected by low-cost sensors which we seek to answer in this work by examining **data collected by more than sixty Real-time Affordable Multi-Pollutant (RAMP) monitors over periods ranging up to eighteen months** in the city of Pittsburgh, PA, USA. First, although various models have been applied to perform calibrations in different contexts, a thorough comparison on a common set of data of several



different forms of calibration models applied to multi-pollutant measurements has yet to be performed. We seek to provide such a comparison and thereby draw robust conclusions about which calibration approaches work best overall and in specific contexts. Second, in previous work with the RAMP monitors and in work with other sensors, unique models have been developed for each sensor. This requires that extensive collocation data be collected for each low-cost sensor, which may not

be feasible if large sensor networks are to be deployed. Therefore, it is important to investigate how well a single generalized calibration model can perform when applied across different individual sensors. Third, it is important to quantify the generalizability of models calibrated using data collected at a specific location to other locations across the same city where the sensors might be deployed, which may not share the same ratios of pollutants. This question is examined with several RAMPs that are co-located with regulatory monitors in the city of Pittsburgh, PA, USA. Finally, we seek to address the stability

of calibration models over time by tracking changes in performance over the course of a year, and from one year to the next. These results will help future deployment efforts for RAMP or similar lower-cost air quality monitors.

## 2 Methods

### 2.1 The RAMP Monitor

The RAMP monitor (Fig. 1) was jointly developed by the Center for Atmospheric Particle Studies at Carnegie Mellon

University (CMU) and a private company, SenSevere (Pittsburgh, PA). The RAMP package combines a power supply, control circuitry, cellular network communications capability, a memory card for data storage, and up to five gas sensors in a weatherproof enclosure. All RAMPs incorporate a nondispersive infrared (NDIR) $CO_2$ sensor produced by SST Sensing (UK), which also measures temperature and relative humidity. All RAMPs have one sensor that measures CO and one sensor that measures $NO_2$. Of the remaining sensors, one is either an $SO_2$ or NO sensor, and the other measures either a combination of

oxidants (referred to hereafter as an Ozone or $O_3$ sensor, since this is its primary function in the RAMP) or Volatile Organic Compounds (VOCs). The VOC sensor is an AlphaSense (UK) PID and all other unspecified sensors are AlphaSense B4 electrochemical units. Further details of the RAMP are provided elsewhere (Zimmerman et al., 2018). **Data collected from a total of 68 RAMP monitors are considered in this work.**

### 2.2 Calibration Data Collection

Following Zimmerman et al. (2018), RAMP monitors are deployed outdoors on a parking lot located on the CMU campus for a calibration based on collocated monitoring with regulatory-grade instruments. The parking lot (40°26'31"N by 79°56'33"W) is a narrow strip between a low-rise academic building to the south and several tennis courts to the north. RAMP monitors are deployed for one month or more to allow for exposure to a wide range of environmental conditions; in 2017, these deployments took place in the summer and fall. Less than 10 meters from the RAMP monitors, a suite of high-quality regulatory-grade

instruments, measuring ambient concentrations of CO, $CO_2$, $O_3$, NO, $NO_2$, $SO_2$, and VOCs (specifically benzene, toluene, ethylbenzene, and xylenes, or BTEX), are stationed to provide true concentration values for these various gases to which the





RAMP monitors are exposed. Using sensor signal data collected by the RAMPs during this collocation period together with data collected by these regulatory-grade instruments, calibration models are created for each RAMP monitor prior to its deployment, as described in Sect. 2.3. Further details on the regulatory-grade instrumentation and the collocation process are provided in previous work (Zimmerman et al., 2018).

In addition to collocation at the CMU campus, additional special collocation deployments of RAMP monitors were performed, in order to allow independent comparisons between the RAMP monitor data and regulatory monitors at different locations. One RAMP monitor was collocated with ACHD regulatory monitors at their Lawrenceville site (40°27'56"N by 79°57'39"W), an urban background site where all NAAQS criteria pollutant concentrations are measured. The ACHD Parkway East site, located alongside the I-376 highway (40°26'15"N by 79°51'49"W), was chosen as an additional collocation site for observing

higher levels of NO and $NO_2$: up to ~100 ppb for NO and ~40 ppb for $NO_2$. For reference, the NAAQS limit for one-year average $NO_2$ is 53 ppb (https://www.epa.gov/criteria-air-pollutants/naaqs-table).

**2.3 Gas Sensor Calibration Models**

Various computational models were applied to the sensor readings of the RAMPs (i.e. the net signal, or raw response minus reference signal, from each electrochemical gas sensor, together with the outputs of the $CO_2$, temperature, and humidity sensor)

to estimate gas concentrations, based entirely on ambient collocations of the RAMPs with regulatory-grade monitors. These models, outlined in the following subsections, include linear and quadratic regression models, a Gaussian process regression model, a nearest-neighbor clustering algorithm, an artificial neural network algorithm, random forest models, and hybrid random forest and linear models.

Models using each of these algorithms were calibrated in three separate categories. First, **individualized RAMP calibration**

**models (iRAMP)** were created for each RAMP, using only the data collected by gas sensors in that RAMP and the regulatory monitors. Individualized models are applied only to data from the RAMP on which they were trained. Second, from these individualized models, a **best individual calibration model (bRAMP)** was chosen, which performed best out of all the individualized models on a testing data set. This model was then used to correct data from all other RAMPs which shared the same mix of gas sensors (to ensure that the inputs to the model would be consistent). Third, **general calibration models**

**(gRAMP)** were developed by taking the median of the data from a subset of the RAMP monitors deployed at the same place and time and treating this as a virtual "typical RAMP", for which models were calibrated for each gas sensor (the median is used rather than the mean to reduce the effects of any erroneous measurements by a few gas sensors in some RAMP monitors on the "typical" signal). Data from about three quarters of the RAMP monitors (53 out of 68) were used for developing the general calibration models (although not all of these monitors were active at the same time). Data from the remaining 15

RAMP monitors were used for testing. In the case of general models, the set of possible model inputs was restricted to ensure that, for each gas, all necessary model inputs would be provided by every RAMP (e.g. for NO models, only CO, NO, $NO_2$, T, and RH could be used as inputs since all RAMP monitors measuring NO would also measure these, but not necessarily any of



the other gases). Thus, each of the calibration model algorithms were applied in three categories, yielding iRAMP, bRAMP, and gRAMP variants of each model.

In all cases, models were calibrated using training data, which consists of the RAMP monitor data collected during the collocation period (which are measurements of the input variables, i.e. the signals from the various gas sensors) together with

the readings of the regulatory-grade instruments with which the RAMP monitor was collocated (which are the targets for the output variables). These collocation data are down-averaged from their original sampling rates to 15-minute averages, to ensure stability of the trained models and minimize the effects of noise on the training process. From the collocation data, eight equally sized, equally spaced time intervals are selected to serve as training data for the calibration models. The cumulative amount of training data is 80% of the collocation data or four weeks of data (corresponding to 2688 15-minute-averaged data points),

whichever is smaller. The minimum amount of training data is 21 days; if less than this is available, no iRAMP model is trained for this RAMP (although bRAMP and gRAMP models trained on other RAMPs are still applied to this RAMP for testing). Any remaining data from the collocation period are left aside as a separate testing set, on which the performance of the trained models is evaluated.

### 2.3.1 Linear and Quadratic Regression Models

Linear regression models represent perhaps the simplest and most common method for gas sensor calibration, and have been used extensively in prior work (Spinelle et al., 2013, 2015; Zimmerman et al., 2018). A linear regression model (sometimes called a multi-linear regression model in the case that there are multiple inputs) describes the output as an affine function of the inputs. Here, linear functions are used where the sets of inputs are restricted to the signal of the sensor for the gas in question along with temperature and relative humidity. For example, the calibrated measurement of CO from the RAMP, $c_{\mathrm{CO}}$,

is an affine function of the signal of the CO sensor, $s_{\mathrm{CO}}$, and the temperature $T$ and relative humidity $RH$ measured by the RAMP:

$$c_{\mathrm{CO}} = \alpha_{\mathrm{CO}}s_{\mathrm{CO}} + \alpha_{\mathrm{T}}T + \alpha_{\mathrm{RH}}RH + \beta_{CO}, \tag{1}$$

Coefficients $\alpha_{\mathrm{CO}}$, $\alpha_{\mathrm{T}}$, and $\alpha_{\mathrm{RH}}$ and offset term $\beta_{CO}$ are calibrated from training data to minimize the root-mean-square difference of $c_{\mathrm{CO}}$ and the measured CO concentration from the regulatory-grade instrument. The one exception to this general

formulation is for evaluation of $c_{O_3}$, where both $s_{O_3}$ and $s_{NO_2}$ are used as inputs (along with $T$ and $RH$); this is done to account for the fact that the sensor for O₃ also responds to NO₂ concentrations (Afshar-Mohajer et al., 2018).

In addition to linear regressions, quadratic regressions were also applied. These are the same as linear regressions but can involve second-order interactions of the input variables. For example, for CO, a quadratic regression function would be of the following form:

$c_{\mathrm{CO}} = \alpha_{\mathrm{CO}}s_{\mathrm{CO}} + \alpha_{\mathrm{CO^2}}s_{\mathrm{CO}}^2 + \alpha_{\mathrm{T}}T + \alpha_{\mathrm{T^2}}T^2 + \alpha_{\mathrm{RH}}RH + \alpha_{\mathrm{RH^2}}RH^2 + \alpha_{\mathrm{CO,T}}s_{\mathrm{CO}}T + \alpha_{\mathrm{CO,RH}}s_{\mathrm{CO}}RH + \alpha_{\mathrm{T,RH}}T\,RH + \beta_{CO},\ (2)$

Note that, as above, a reduced set of inputs is used here. Quadratic regression models using such reduced sets (the same sets used for linear regression) are hereafter referred to as "limited" quadratic regression models; in contrast, models making full



use of all available gas, temperature, and humidity sensor inputs from a given RAMP are referred to as "complete" quadratic regression models.

The main advantages of linear and quadratic regression models are their ease of implementation and calibration, as well as their ability to be readily interpreted, e.g., the relative magnitudes of the regression coefficients correspond to the relative importance of the different inputs in producing the output. The main disadvantage of these models is their inability to compute complicated relationships between input and output which are beyond that of a second-order polynomial. The training and application of linear and quadratic regression models are implemented using custom-written routines for the MATLAB programming language (version R2016b).

### 2.3.2 Gaussian Process Models

Gaussian processes are a form of regression which generalizes the multivariate Gaussian distribution to infinite dimensionality (Rasmussen and Williams, 2006). For the purposes of calibration, we make use of a simplified variant of a Gaussian process model. From the training data, both the signals of the RAMP monitors and the readings of the regulatory-grade instruments are transformed such that their distributions during the training period can be approximately modelled as standard normal distributions. This transformation is accomplished by means of a piecewise linear transformation, where the domain is segmented and for each segment different linear mappings are applied. After this transformation, an empirical mean vector $\boldsymbol{\mu}$ and covariance matrix $\boldsymbol{\Sigma}$ is computed for the regulatory-grade and RAMP measurements. The transformed measurements can then be described using a multivariate Gaussian distribution. For example, for a RAMP measuring CO, $SO_2$, $NO_2$, $O_3$, and $CO_2$, this distribution would be:

$$\left\{c'_{CO}, c'_{SO_2}, c'_{NO_2}, c'_{O_3}, c'_{CO_2}, s'_{CO}, s'_{SO_2}, s'_{NO_2}, s'_{O_3}, s'_{CO_2}, T', RH'\right\}^{\mathrm{T}} \sim \mathcal{N}(\boldsymbol{\mu}, \boldsymbol{\Sigma}), \tag{3}$$

where, for example, $c'_{CO}$ represents the concentration measurement for CO following the transformation. The mean vector and covariance matrix are divided as follows:

$$\boldsymbol{\mu} = \begin{bmatrix} \boldsymbol{\mu}_{conc} \\ \boldsymbol{\mu}_{RAMP} \end{bmatrix} \quad \boldsymbol{\Sigma} = \begin{bmatrix} \boldsymbol{\Sigma}_{conc,conc} & \boldsymbol{\Sigma}_{conc,RAMP} \\ \boldsymbol{\Sigma}^{\mathrm{T}}_{conc,RAMP} & \boldsymbol{\Sigma}_{RAMP,RAMP} \end{bmatrix}, \tag{4}$$

where $\boldsymbol{\mu}_{conc}$ represents the mean of the (transformed) concentration measurements of the regulatory-grade instrument, $\boldsymbol{\mu}_{RAMP}$ represents the mean of the (transformed) signal measurements from the RAMP, $\boldsymbol{\Sigma}_{conc,conc}$ represents the covariance of the (transformed) concentrations, $\boldsymbol{\Sigma}_{RAMP,RAMP}$ represents and covariance of the (transformed) RAMP signals, and $\boldsymbol{\Sigma}_{conc,RAMP}$ represents the covariance between the (transformed) concentrations and RAMP signals ($\boldsymbol{\Sigma}^{\mathrm{T}}_{conc,RAMP}$ is the transpose of $\boldsymbol{\Sigma}_{conc,RAMP}$). Once these vectors and matrices have been defined, the model is calibrated.

Given a new set of signal measurements from a RAMP, denoted as $\mathbf{y}_{RAMP} = \left\{s_{CO}, s_{SO_2}, s_{NO_2}, s_{O_3}, s_{CO_2}, T, RH\right\}^{\mathrm{T}}$, these are transformed using the piecewise linear transformation defined above to give the set of transformed signal measures $\mathbf{y}'_{RAMP}$. These are then used to estimate the concentrations measured by the RAMP with the standard conditional updating formula of the multivariate Gaussian as follows:





$$\{c'_{CO}, c'_{SO_2}, c'_{NO_2}, c'_{O_3}, c'_{CO_2}\}^{T} = \mathbf{\mu}_{conc} + \mathbf{\Sigma}_{conc,RAMP}\mathbf{\Sigma}^{-1}_{RAMP,RAMP}(\mathbf{y}'_{RAMP} - \mathbf{\mu}_{RAMP}), \quad\quad\quad (5)$$

The inverse of the original piecewise linear transformation is then applied to these transformed concentration estimates to yield the appropriate concentration estimates in their original units.

The main advantage of a Gaussian process calibration model of this form is its robustness to incomplete or inaccurate

information; for example, if a signal from one gas sensor were missing or corrupted by a large voltage spike, in the former case the missing input could be "filled in" by the correlated measurements of other sensors, while in the latter case estimates would be "reigned in" by the more reasonable measures of the other sensors. A major disadvantage of this calibration model is its continued use of what is basically a linear regression formula; the only difference being in the non-linear transformation from the original measurement space to the standard normal variable space used by the model. Furthermore, during the

calibration process, the ratios of concentration for the pollutants of the collocation site may be "learned" by the model, making it less likely to predict differing ratios during field deployment. The training and application of Gaussian process calibration models are accomplished using custom-written routines in the MATLAB programming language.

### 2.3.3 Clustering Model

Clustering models, also referred to as nearest neighbor models, seek to estimate the outputs corresponding to new inputs by

searching for input-output pairs in the training data for which the distance (by a predefined distance metric in a potentially high-dimensional space) between the new input and the training inputs is minimized, and using the average of several outputs corresponding to these nearby inputs (the "nearest neighbors"). To prevent the need to store every input-output pair from the training data for comparison to new inputs, the input data are first clustered, i.e., grouped by proximity of the input data. These clusters are then represented by their centroid, with the corresponding output being the mean of the outputs from the clustered

inputs. In this work, training data are grouped into one thousand clusters using the 'kmeans' function in MATLAB. Euclidian distance in the multidimensional space of the sensor signals from each RAMP is used. For estimation, the outputs of the five nearest neighbors to a new input are averaged. A similar calibration method has been used in previous work (Hagan et al., 2017).

A major advantage of this approach is its simplicity and flexibility, allowing it to capture complicated nonlinear input-output

relationships by referring to past records of these relationships, rather than attempting to determine the actual pattern which these relationships follow. Such a method can perform very well when the relationships are stable, and when any new input with which the model is presented is similar to at least one of the inputs from the training period. However, generalizing beyond the training period is difficult, and the model will tend to perform poorly if the "nearest neighbors" of a new input are in fact quite far away, in terms of the distance metric used, from this input. This is of potential concern for air quality

applications, as the detection of high concentrations is an important consideration. If no high concentrations are observed during the collocation period, then the resulting trained model will be unable to estimate such high concentrations if it is





exposed to these during deployment. This difficulty in generalizing beyond the training data set is in fact a common difficulty for many nonparametric models, of which the clustering model is an example.

### 2.3.4 Artificial Neural Network Model

The artificial neural network model, or simply neural network, is a machine learning paradigm which seeks to replicate, in a
simplified manner, the functioning of an animal brain in order to perform tasks in pattern recognition and classification (Aleksander and Morton, 1995). A basic neural network consists of several successive layers of "neurons". These neurons each receive a weighted combination of inputs from a higher layer (or the signal inputs, if they are in the top layer) and apply a simple but nonlinear function to them, producing a single output which is then fed on into the next layer. By including a variety of possible functions performed by the neurons and appropriately tuning the weights applied to inputs fed from one
layer to the next, highly complicated nonlinear transformations can be performed in successive small steps.

Neural networks have been applied to a large number of problems, including the calibration of low-cost gas sensors (Spinelle et al., 2015). Neural networks represent an extremely versatile framework, and are able to capture nearly any nonlinear input-output relationship (Hornik, 1991). Unfortunately, to do so may require vast amounts of training data, which it is not always practical to obtain. Calibration of these models is also a time-consuming process, requiring many iterations to tune the
weightings applied to values passed from one layer to the next. In this work, neural networks were trained and applied using the 'Netlab' toolbox for MATLAB (Nabney, 2002). The network has a single hidden layer with twenty nodes. To limit the computation time needed for model training, the number of allowable iterations of the training algorithm was capped at ten thousand; this cap was typically reached during the training.

### 2.3.5 Random Forest and Hybrid Models

A random forest model is a machine learning method which makes use of a large number of decision "trees". These trees are hierarchical sets of rules which group input variables based on thresholding (e.g. "the third input variable is above or below a given value"). The thresholds used for these rules as well as the inputs they are applied to and the order in which they are applied are calibrated during training. The final groupings of input variables from the training data, located at the end or "leaves" of the branching decision tree, are then associated with the mean values of the output variables for this group (similar
to a clustering model). For estimating an output given a new set of inputs, each decision tree within the random forest applies its sequence of rules to assign the new data to a specific "leaf", and outputs the value associated with that leaf. The output of the random forest is the average of the outputs of each of its trees.

A primary shortcoming of the random forest model (which it shares with other nonparametric methods) is its inability to generalize beyond the range of the training data set, i.e., outputs of a random forest model for new data can only be within the
range of the values included as part of the training data. For this reason, the standard random forest model was also expanded into a hybrid random forest and linear regression model as suggested by Zimmerman et al. (2018) and similar to the approach of Hagan et al. (2017) who combine nearest neighbor and linear regression models in a similar fashion. In this modified model,





a random forest is applied to new data to estimate the concentrations of various measured pollutants. For example, the concentration of CO measured by a RAMP including sensors for CO, $SO_2$, $NO_2$, $O_3$, and $CO_2$ is estimated using a random forest as:

$$c_{\text{CO}} = \text{RF}_{CO}\left(s_{CO}, s_{SO_2}, s_{NO_2}, s_{O_3}, s_{CO_2}, T, RH\right), \tag{6}$$

If this estimated concentration exceeds a given value (in this case, 90% of the maximum concentration value observed during the training, corresponding to about 1ppm in the case of CO), a linear model of the form of Eq. (1) is instead used to estimate the concentration. This linear model is calibrated using a 15% subset of the training data with the highest concentrations of the target gas and is therefore better able to extrapolate beyond the upper concentration value observed during the training period. This hybrid model therefore is designed to combine the strengths of the random forest model, i.e. its ability to capture

complicated nonlinear relationships between various inputs and the target output, with the ability of a simple linear model to extrapolate beyond the set of data on which the model is trained. Random forests are implemented using the 'TreeBagger' function in MATLAB, and custom routines are used to implement hybrid models.

## 2.4 Assessment Metrics

In the following section, the performance of the calibration models in translating sensor signals to concentration estimates is

assessed in several ways. The estimation bias is assessed as the mean normalized bias (MNB), the average difference between the estimated and actual values, divided by the mean of the actual values. That is, for $n$ measurements:

$$\text{MNB} = \frac{\sum_{i=1}^{n}(c_{\text{estimated},i} - c_{\text{true},i})}{\sum_{i=1}^{n}(c_{\text{true},i})}, \tag{7}$$

where $c_{\text{estimated},i}$ is the measured concentration as estimated by the RAMP monitor and $c_{\text{true},i}$ is the corresponding true value measured by a regulatory-grade instrument. The variance of the estimation is assessed via the coefficient of variation of the

mean absolute error (CvMAE), the average of the absolute differences between the estimated and actual values divided by the mean of the actual values. The estimates used in evaluating the CvMAE are corrected for any bias as determined above:

$$\text{CvMAE} = \frac{\sum_{i=1}^{n}|c_{\text{estimated},i} - n_{\text{bias}} - c_{\text{true},i}|}{\sum_{i=1}^{n}(c_{\text{true},i})}, \tag{8}$$

where:

$$n_{\text{bias}} = \frac{1}{n}\sum_{i=1}^{n}\left(c_{\text{estimated},i} - c_{\text{true},i}\right), \tag{9}$$

Correlation between estimated and actual concentrations is assessed using the Pearson linear correlation coefficient (r):

$$r = \frac{\sum_{i=1}^{n}\left(c_{\text{estimated},i} - \frac{1}{n}\sum_{j=1}^{n}c_{\text{estimated},j}\right)\left(c_{\text{true},i} - \frac{1}{n}\sum_{j=1}^{n}c_{\text{true},j}\right)}{\sqrt{\sum_{i=1}^{n}\left(c_{\text{estimated},i} - \frac{1}{n}\sum_{j=1}^{n}c_{\text{estimated},j}\right)^2}\sqrt{\sum_{i=1}^{n}\left(c_{\text{true},i} - \frac{1}{n}\sum_{j=1}^{n}c_{\text{true},j}\right)^2}}, \tag{10}$$

Intuitively, these basic metrics are used to quantify the difference in averages between estimated and true concentrations (MNB), the average of differences between these (CvMAE), and the similarity in their behavior (r).

In addition to the above metrics, EPA methods for evaluating precision and bias errors are used as outlined in Camalier et al.

(2007). To summarize, the precision error is evaluated as:



$$\text{Precision} = \sqrt{\frac{n\sum_{i=1}^{n}\delta_i^2-\left(\sum_{i=1}^{n}\delta_i\right)^2}{n\chi_{0.1,n-1}^2}}, \tag{11}$$

where $\chi_{0.1,n-1}^2$ denotes the $10^{\text{th}}$ percentile of the chi-squared distribution with $n-1$ degrees of freedom and the percent difference in the $i^{\text{th}}$ measurement is evaluated as:

$$\delta_i = \frac{c_{\text{estimated},i}-c_{\text{true},i}}{c_{\text{true},i}}\cdot 100, \tag{12}$$

The bias error is computed as:

$$\text{Bias} = \frac{1}{n}\sum_{i=1}^{n}|\delta_i| + \frac{t_{0.95,n-1}}{n}\sqrt{\frac{n\sum_{i=1}^{n}\delta_i^2-\left(\sum_{i=1}^{n}|\delta_i|\right)^2}{n-1}}, \tag{13}$$

where $t_{0.95,n-1}$ is the $95^{\text{th}}$ percentile of the t distribution with $n-1$ degrees of freedom. Prior to the computation of precision and bias, measurements where the corresponding true value is below an assigned lower limit are removed from the measurement set to be evaluated, so as not to allow near-zero denominator values in Eq. (12). Lower limits used in this work

are based on the guidelines presented by Williams et al. (2014) and are listed in Table 1.

Using the EPA precision and bias calculations allows for these values to be compared against performance guidelines for various sensing applications, as presented Williams et al. (2014) and listed in Table 2. For the RAMP monitors, a primary goal is to achieve data quality sufficient for hotspot identification and characterization (Tier II) or personal exposure monitoring (Tier IV), which requires that both precision and error bias metrics be below 30%. A supplemental goal is to achieve

performance sufficient for supplemental monitoring (Tier III), requiring precision and bias metrics below 20%.

## 3 Results

In this section, we examine the performance of the RAMP gas sensors and the various calibration models applied to their data. We will focus our attention on the CO, NO, $NO_2$, and $O_3$ sensors. Calibration of measurements by the $SO_2$ and VOC sensors in an urban environment, unlike the near-source $SO_2$ calibration work of Hagan et al. (2017), is ongoing and will be presented

in forthcoming manuscripts. Results for calibration of measurements by the $CO_2$ sensors are presented in supplemental figures.

### 3.1 Performance across Individualized Models on CMU Site Collocation Data

Figure 2 presents a comparison of the performance of various calibration models applied to testing data collected at the CMU site during 2017. As described in Sect. 2.3, collocation data are divided into training and testing sets, with the former (always being between three and four weeks in total duration) used for model development and the latter used to test the developed

model using the assessment metrics described in Sect. 2.4, as presented in Fig. 2. All models in the figure are of the "iRAMP" category, being developed using only data collected by a single RAMP and the collocated regulatory-grade instruments. In the figure, squares indicate the median performance across all RAMPs for each performance metric, and the error bars span from the $25^{\text{th}}$ to $75^{\text{th}}$ percentiles of each metric across the RAMPs. **For CO, 48 iRAMP models are compared; for NO, 19 models;**





**for NO₂, 62 models; for O₃, 44 models**. Note that only 20 RAMP monitors included an NO sensor. An iRAMP model was not developed for RAMP monitors that had fewer than 21 days of collocation data with the relevant regulatory-grade instrument. The figures are arranged such that the lower-left corner denotes "better" performance (CvMAE close to 0 and r close to 1).

Performance of the calibration models varies with the type of gas sensor being calibrated, as well as with the calibration model used. For CO, performance is consistent across most models, with the quadratic and linear regression models, clustering, and random forest models having r typically above 0.8 and CvMAE typically below 0.25. Bias for these CO models is also low, typically being within ±10 ppb (MNB less than 1%). Neural networks and Gaussian process models show worse performance for CO, with the spread in the performance metrics also being large between RAMPs. In the case of NO₂, however, neural

networks perform similarly to random forest, hybrid, clustering, and complete quadratic regression, all of which perform better than limited quadratic models, linear models, and Gaussian process models. Bias for these NO₂ models is in the range of ±2 ppb (MNB less than 20%). For O₃, similar to what was observed for CO, most models have similar performance, with only clustering and Gaussian process models having typical r below 0.8 and CvMAE above 0.25. Bias for these O₃ models is low, in the range of ±1 ppb (MNB less than 2%). Finally, for NO, most models have r ranging from 0.6 to 0.9 and CvMAE ranging

from 0.5 to 2. Bias for NO is in the range of ±2 ppb (MNB less than 10%).

To summarize, for CO and O₃, the simple parametric complete quadratic regression models perform better than other more complicated modelling approaches, and even linear regression models give reasonable results. For NO₂ and NO, while more complicated random forest or neural network models perform best, complete quadratic regression models can give comparable performance. Quadratic regression, random forest, and hybrid random forest and linear regression models gives the most

consistent performance, being among the top four methods across all gases. Table 3 lists the modelling methods which meet the EPA performance guidelines (Table 2) based on performance at the CMU site in 2017. All methods meet at least Tier I (educational monitoring) criteria for all gases considered. Most methods fall within the Tier II (hotspot detection) or Tier IV (personal exposure) performance levels for all gases. For CO, random forest and quadratic regression methods meet Tier III (supplemental monitoring) criteria. Finally, in Table 4, additional information is presented about these performance results,

including measured concentration ranges during training and testing periods, durations of these periods, and additional performance metrics including un-normalized MAE and bias in the measured concentration units, to allow for direct comparison with the concentrations ranges.

**3.2 Comparison of Individualized, Best, and General Models on CMU Site Collocation Data**

Next, we examine how the performance of the best individual models (bRAMP) and of the general models (gRAMP) applied

to all RAMPs compare to the performance of the individualized RAMP (iRAMP) models presented in the last section. For simplicity, we restrict ourselves to three models for each gas, chosen from among the better-performing iRAMP models and including at least one parametric and one non-parametric approach. Figure 3 presents these comparisons.



For CO, limited quadratic regression, complete quadratic regression, and random forest models are compared. While individualized models all perform relatively well, the limited quadratic model is the most amenable to generalization, with the gRAMP version performing nearly as well as the iRAMP version. This is to be expected, as the limited quadratic regression model has a simple parametric form and is thus the least susceptible of the models shown to overfitting during training. For

$O_3$, complete and limited quadratic regression and hybrid random forest/linear regression models are compared. Here, bRAMP and gRAMP models have fairly similar performance (median r between 0.8 and 0.9, median CvMAE between 0.2 and 0.25), which is only slightly worse than the iRAMP models (where median r is just above 0.9 and median CvMAE is just below 0.2). The spread in model performance when moving from iRAMP to gRAMP models is smallest for the $O_3$ hybrid models. For NO, linear regression, neural network, and hybrid random forest/linear regression models are compared. While iRAMP and

bRAMP hybrid and neural network models show similar performance for NO (median r between 0.7 and 0.8, median CvMAE below 1.5), generalization of the linear models is poorer. NO gRAMP models have higher CvMAE than their iRAMP versions, although the neural network model is the least affected. For $NO_2$, linear, random forest, and neural network models are compared. The random forest and neural network models maintain a fairly close grouping (median r and CvMAE differing by less than 0.05), while the complete quadratic models perform less well in their bRAMP and gRAMP versions.

Across all gases and models, iRAMP models tend to perform best, as might be expected since these models are both trained and applied to data collected by a single RAMP monitor, and therefore will account for any peculiarities of individual sensors. Between the bRAMP models, in which a model is trained using data from a single RAMP and applied across multiple RAMPs, and gRAMP models, which are trained on data from a virtual "typical" RAMP (composed of the median signal from several RAMPs) and then applied across other RAMPs, it is difficult to say which approach would be better, as results vary by gas as

well as by modelling approach. For parametric models (i.e., linear and quadratic regression) the bRAMP and gRAMP versions typically have similar performance. For non-parametric models (i.e., neural network, random forest, and hybrid models), performance of bRAMP versions is typically better than the gRAMP versions, although in the case of $NO_2$ and $O_3$ the performance is comparable.

### 3.3 Performance of Selected Models at Regulatory Monitoring Sites

Figure 4 depicts the performance of calibration models for RAMP monitors deployed at two EPA monitoring stations operated by the ACHD. Fully colored points indicate the performance of the models at these sites, while hollow points indicate the performance of the corresponding RAMP at the CMU site for comparison. For each gas type, different calibration models are used, chosen from among the models depicted in Fig. 3. Models trained at the CMU site (as presented in previous sections) are used to calibrate data collected by the RAMP monitor at the station. Also note that not all gases monitored by RAMPs are

monitored by the stations, hence why only one station may appear in each plot.
For CO at the Lawrenceville site, performance of the gRAMP model is better in terms of correlation (0.8 vs. 0.7) and nearly the same in terms of CvMAE (0.35 vs. 0.3) than the iRAMP model, but both models perform worse than at the CMU site (where r was higher by between 0.05 and 0.2 and CvMAE was lower by about half). For $O_3$, both iRAMP and gRAMP models



have very similar performance at the Lawrenceville site compared to the CMU site, with r being nearly the same (above 0.9) and CvMAE being slightly (less than 0.05) worse. Concentrations of $O_3$ are comparable at both sites (annual mean ~25 ppb, standard deviation ~15 ppb). For $NO_2$, both iRAMP and gRAMP models perform worse than at CMU (CvMAE is about 0.1 higher for both models, and r is 0.05 lower for the iRAMP model), but the gap in performance is smaller for the gRAMP model

since its correlation is nearly the same. The overall performance of the iRAMP and gRAMP models for this gas at Lawrenceville are comparable (r about 0.8, CvMAE about 0.6).

For Parkway East, both iRAMP and gRAMP CO models have better performance than at the CMU site with respect to correlation (with r above 0.95), but for the gRAMP model performance with respect to CvMAE is worse (CvMAE is greater than 0.4, as opposed to being less than 0.2 for the iRAMP model). It should be noted that both the average concentration and

variability in concentration (as measured by the standard deviation) of CO are more than double at the Parkway East site compared to either the CMU or Lawrenceville sites. For NO, the iRAMP model performs slightly worse in terms of correlation (r of 0.69 vs. 0.71) at the Parkway East site than at the CMU site, while the gRAMP model performs better (r of 0.78). In fact, the performance of the gRAMP model at Parkway East is better than that of the iRAMP model at CMU. Levels of NO are about 50% higher at Parkway East than at CMU (6 vs. 4 ppb annual mean in 2017), but variability is about 25% less. Finally,

for $NO_2$, as for the Lawrenceville site, performance of both iRAMP and gRAMP models at Parkway East is comparable (r about 0.7, CvMAE about 0.45). However, it should be noted that, while performance for the iRAMP model is worse than at the CMU site, performance of the gRAMP model is better.

To evaluate the performance of these sensors in a different way, EPA-style precision and bias metrics are provided in Fig. 5. Only CO, $O_3$, and $NO_2$ are considered, as these are the gases for which performance guidelines have been suggested by the US

EPA (Table 2). These guidelines are indicated by the dotted boxes in the figure; points falling within the box meet the criteria for the corresponding tier. Also, the range in observed performance at the CMU site, as depicted in Fig. 3, is reproduced here for comparison using black markers with error bars. For CO, by these criteria, the gRAMP model outperforms the iRAMP model, with the gRAMP model meeting Tier II or IV criteria for all locations, while for the iRAMP model, Tier III criteria are met at the CMU site but only Tier I criteria are met at the other sites. Thus, under these metrics, for CO the gRAMP model is

more representative of performance at other sites, while for the iRAMP model performance is more varied between sites. For $O_3$, performance of both models at Lawrenceville is better than assessed at the CMU site, as seen before under the r and CvMAE metrics. Both models fall near the boundary between Tiers II/IV and Tier III performance criteria. For $NO_2$, performance at ACHD sites in terms of the bias is always worse than predicted by the CMU performance, although in terms of precision, the gRAMP model at the CMU site better represents the AHCD site performance than the iRAMP model (the

same trend is seen under the r and CvMAE metrics for $NO_2$).

Overall, there tends to be a change in model performance at either of the deployment sites as compared to the CMU site. This is to be expected to some degree, as the concentration range and mixture of gases (especially at the Parkway East site, which is located next to a major highway) can be different at a new site (where the model was not trained), and thus cross-sensitivities of the sensors may be affected. These differences appear to be greatest for CO, with performance being *better* at the Parkway



East site, where overall CO concentrations are higher. Additionally, gRAMP models tend to perform as well as or better than iRAMP models when monitors are deployed to new sites, and the performance of the gRAMP models at the training site is typically more representative of the expected performance at other sites than that of the iRAMP models. This is likely because, while the iRAMP models are trained for individual RAMPs at the training site, the gRAMP models are trained across multiple

RAMPs at that site, and therefore are more robust to a range of different responses for the same atmospheric conditions. Thus, when a RAMP is moved from one site to another, and its responses change slightly due to a change in the surrounding conditions, the gRAMP model will be more robust against these changes.

**3.4 Performance of Calibration Models over Time**

We now examine the change in performance of calibration models over time. Figure 6 shows the performance of models
developed based on data collected at the CMU site in both 2016 and 2017 and tested on data collected in these two years. Thus, any change in performance between these two models will indicate the degree to which the models have changed from one year to the next. Note that NO is omitted here because data to build calibration models for this gas were not collected in 2016. Also note that results presented in previous sections have only used data collected in 2017 for model training and evaluation. For CO, there is a larger change in performance between 2016 and 2017 models when tested on 2016 data as opposed to when
they are tested on 2017 data, with performance of 2017 models on 2016 data being worse (mostly in terms of CvMAE). Performance of models trained in 2016 on data collected in the same year is also better than that of 2017 models on the data from 2017. However, except for the 2017 gRAMP models applied to 2016 data (which could indicate a change in sensor response from as-new to in-use condition), in the three other cases CvMAE is typically under 0.25 and Pearson r over 0.9. For $O_3$, 2016 models applied to 2016 data again outperform 2017 models applied to 2017 data, but in this case the performance
ranges of the models overlap from one year to the next, indicating minimal degradation in performance. For $NO_2$, there is a decrease in performance from 2016 to 2017, with r decreasing by about 0.05 to 0.1 and CvMAE increasing by about 0.1 to 0.2.

The drop in performance when models from one year are applied to data collected in the next year is relatively consistent for all gases and all modelling approaches depicted. This suggests that degradation is occurring in the sensors, reducing the
intensity of their responses to the same ambient conditions and/or changing the relationships between their responses, such that a model calibrated on the response characteristics of the sensors in one year will not necessarily perform as well using data collected by the same sensors in a different year. This is consistent with the operation of the electrochemical sensors, where electrode material is used up over time as part of the normal functioning of the sensor. To compensate for this, new models should be calibrated for sensors on at least an annual basis, to keep track with changes in signal response. Furthermore,
calibration models should preferably be applied to data from sensors with a similar age to avoid effects due to different signal responses of sensors which have degraded to varying degrees. Finally, in comparing model performance from one year to the next, there is no significant increase in error associated with using gRAMP models (trained for sensors of a similar age) rather than iRAMP or bRAMP models.





### 3.5 Changes in Field Performance over Time

Finally, we track the performance of RAMPs over time at specific deployment locations, as depicted in Fig. 7, to evaluate changes in calibration model field performance over time. This is done using three RAMP monitors; one was deployed at the ACHD Lawrenceville station from January through September of 2017, as well as during November and December. A second

RAMP was kept at the CMU site year-round, where it was collocated with regulatory-grade instruments intermittently between May and October. The third RAMP was deployed at the ACHD Parkway East site beginning in November of 2017. The same calibration models as depicted in Fig. 4 (using the training data collected at the CMU site in 2017) are used. Performance of CO, $NO_2$, and $O_3$ sensors are depicted, as both CO and $NO_2$ were continuously monitored at both of the ACHD sites and intermittently monitored at the CMU site, and $O_3$ was consistently monitored at ACHD Lawrenceville as well as being

intermittently monitored at the CMU site.

For CO, the limited quadratic regression gRAMP model is used, since it demonstrated the best performance of the general models in Sect. 3.2. In terms of CvMAE, performance is consistently good (0.5 or less) throughout the year, with slight differences from week to week. In terms of correlation, CO experiences intermittent periods of low correlation (e.g. July to September), during which performance varies greatly from week to week. There does not appear to be a clear seasonal or

temporal trend in this performance, as low correlations occur in the late summer but not early summer, and there is also a drop in performance in the winter at the Lawrenceville site, possible indicative of long-term degradation. Additionally, periods of lower correlation at the Lawrenceville site do not appear to coincide with periods of atypical CO concentrations, nor with periods of excessive CO variability at this site, nor with any unusual pattern in the other factors measured by the RAMP. Periods of lower performance appear to roughly coincide for the CMU and Lawrenceville sites for the time during which both

sites were active, and observed ranges of CO concentration were comparable for both the CMU and Lawrenceville sites during this period. Correlation is uniformly high at the Parkway East site, where overall CO concentrations are higher, but correlation at the Lawrenceville site was also high for the same period. For $O_3$, the hybrid random forest gRAMP model is used, as it showed the least variability in performance of the gRAMP models in Fig. 3. Overall, performance of the calibrated $O_3$ measurements is good, with almost uniformly high correlation and relatively low CvMAE throughout the year. For $NO_2$, a

random forest gRAMP model is used, which provided the same performance as the neural network gRAMP model in Fig. 3. Although it is higher than for the other gases, CvMAE is fairly consistent throughout the year. Correlation shows a similar trend to that of CO, with correlation becoming lower and more variable during certain periods. Here, these periods are concentrated in the fall. Also, correlation at the Parkway East site is lower than at the Lawrenceville site during the same period. This is potentially due to the higher variability of $NO_2$ observed at the Lawrenceville site (standard deviation, SD 13

ppb) during this period, as compared to the Parkway East site (SD 6 ppb).





## 4 Discussion and Conclusions

Based on the results presented in Sect. 3.1, random forest, complete quadratic regression, and hybrid models give the best and most consistent performance across all gases. Of these, the hybrid models, combining the complicated non-polynomial behaviors of random forest models (capable of capturing unknown sensor cross-sensitivities) with the generalization

performance of parametric linear models, tend to generalize best for NO, $NO_2$, and $O_3$ when applied to data collected at new sites. For CO, quadratic regression models generalize better. Neural networks perform well for NO and $NO_2$ but not for CO; limited quadratic regression models perform well for CO and $O_3$ but not for NO and $NO_2$. Linear regression, Gaussian processes and clustering are the worst overall models for these gases, never being in the top two best performing models, and only rarely being one of the top three. These results could perhaps be improved further; for instance, our linear and quadratic

regression models did not use regularization, nor did we experiment with neural networks involving multiple hidden layers and varying numbers of nodes. The fact that for most gases a variety of calibration approaches show similar and (for typical uses cases, acceptable) performance may reflect better underlying performance from the RAMP monitor, as similar studies for other low-cost sensor packages showed a wider variability in performance between calibration approaches (see e.g. the summary provided by Zimmerman et al., 2018).

Overall, in most cases the generic bRAMP and generalized gRAMP calibration models perform worse than the individualized iRAMP models at the calibration site, but the decline in performance may be manageable and acceptable depending on the use case. For example, for $NO_2$ (Fig. 3), median performance of bRAMP and gRAMP neural network and random forest models are only 15% worse in terms of CvMAE and 5% worse is terms of r. For $O_3$, median performance of all models is above 0.8 for r and below 0.25 for CvMAE, indicating a high level of correlation and relatively low estimation error. For CO, limited

quadratic models meet the same criteria. Furthermore, in examining the generalization performance of the models when applied to new sites, as depicted in Figs. 4 and 5, for $NO_2$ (in terms of the r and CvMAE metrics) and for CO (in terms of the EPA precision and bias metrics), the gRAMP models show more consistent performance between the calibration and deployment locations than the iRAMP models. For $O_3$, performance of iRAMP and gRAMP models at the Lawrenceville site is comparable, while for NO, performance of the gRAMP model at the Parkway East site is actually better than the iRAMP model. This may

indicate that the NO sensors are more affected by changes in ambient conditions than the other electrochemical sensors, and the gRAMP model is better able to average out these sensitivities than the other model categories considered.

Based on comparisons between the performance of models from one year to the next, as well as the analysis of changes in performance of the RAMP monitors collocated with regulatory-grade instruments for long periods, some sensors, such as the $O_3$ sensor, are quite stable over time. For the CO sensor, performance seemed variable over time, and performance noticeably

degraded from one year to the next, although no seasonal trends were apparent, and overall performance may be acceptable (CvMAE < 0.5). The $NO_2$ also exhibited some degradation from one year to the next, although performance was stable over time in 2017, with minimal changes in overall performance during this long deployment period.



It can generally be expected that RAMP monitors will at least meet Tier II or Tier IV EPA performance criteria for $O_3$ (with hybrid linear and random forest bRAMP and possibly gRAMP models) and CO (with limited quadratic regression gRAMP models). Individualized calibration models are more likely to meet Tier II/IV criteria for $NO_2$ (with iRAMP or bRAMP random forest models), and localized calibration may also be required. For NO, while no specific target criteria are established, neural

network iRAMP or gRAMP models appear to perform best.

## 4.1 Recommendations for Future Low-Cost Sensor Deployments

In comparing different methods for the calibration of electrochemical sensor data, it was found that in some cases, e.g. for CO, simple parametric models, such as quadratic functions of a limited subset of the available inputs, were sufficient to transform the signals to concentration estimates with a reasonable degree of accuracy. For other gases, e.g. $NO_2$, more sophisticated

nonparametric models performed better, although parametric quadratic regression models making use of all sensor inputs were still among the best performing models for all gases. Depending on the application, therefore, different methods might be appropriate, e.g., using simpler parametric models such as quadratic regression to calibrate measurements and provide real-time estimates, while using more sophisticated non-parametric methods such as random forest models when performing long-term analysis for exposure studies. Of the non-parametric methods considered, random forest and hybrid random forest and

linear regression models gave the best general performance across all the gas types. These models, along with the quadratic models, should therefore be considered for situations where it is desirable to use the same type of calibration across all gases, e.g. to reduce the "overhead" of programming multiple calibration approaches. The hybrid model, which combines the flexibility of the random forest with the generalizability of the linear model, is most theoretically promising for general application, but in practice the hybrid models give similar performance to random forest models only. This is likely because

in the analysis presented here, training and testing data showed similar concentration ranges (see Table 4), and thus the capabilities of the hybrid model to generalize beyond the range of the training data was not adequately realized. Future work will also investigate other forms of hybrid models. For example, combinations of neural network and linear regression models may work well for NO, where neural networks provided better performance than hybrid models using random forests. Also, for CO and Ozone, hybrid models combining random forests with quadratic models might perform better than those with linear

models, since quadratic models perform better than linear models for these gases overall.

Although there is a reduction in performance as a result of not using individualized monitor calibration (iRAMP) models when these are calibrated and tested at the same location, the use of a single calibration model across multiple monitors, representing either the best of available individualized models (bRAMP) or a general model developed for a "typical" monitor (gRAMP), tends to give more consistent generalization performance when tested at a new site. This suggests that variability in the

responses of individual sensors for the same gas when exposed to the same conditions (such as would be accounted for when developing separate calibration models for each monitor) tends to be lower than the variability in the response of a single sensor when exposed to different ambient environmental conditions and a different mixture of gases (such as is experienced when the monitor is moved to a new site). Models that are developed and/or applied across multiple monitors will avoid



"overfitting" to the specific response characteristics of a single sensor in a single environment. Thus, considering that it is impractical to perform a collocation for each monitor at the location where it is to be deployed, there is little benefit to developing individualized calibration models for each monitor when their performance will be similar to (if not worse than) that of a generalized model when the monitor is moved to another location.

There are several additional qualitative advantages to using gRAMP models. First, the effort required to calibrate models is reduced, since not every monitor needs to be present for collocation and separate models do not have to be created for every monitor. For example, while for CO only 48 RAMPs had sufficient data to calibrate iRAMP models based on data collected at the CMU site in 2017, general models can be calibrated and applied for all 68 RAMPs which were at the CMU site during this period, as well as for additional RAMPs which were never collocated at the CMU site but had the same gas sensors

installed. Second, collocation data collected from multiple RAMP monitors at different sites can be combined in the creation of a gRAMP model, whereas iRAMP or bRAMP models would require each RAMP monitor to be present at each collocation site. This means that a wider range of ambient gas concentrations can be reflected in the training data, allowing for better generalization. Finally, the use of gRAMP models allows for robustness against noise of individual sensors, which can lead to mis-calibration of iRAMP models but is less likely to do so if data from multiple sensors are averaged. Therefore, for future

deployments, generalized models applicable across all monitors should be used.

For long-term deployments, it is recommended that new models be developed each year, due to the noticeable change in performance when models for one year were used for processing data collected in the subsequent year. If generalized models are used, model development can be performed using only a representative subset of monitors, allowing most monitors to remain deployed in the field. Another option is to maintain a few "gold standard" monitors collocated with regulatory-grade

instruments year-round and to use these monitors for the development of generalized models to be used with all field-deployed monitors over the same period. Determination of how many monitors are necessary to develop a sufficiently robust generalized model is a topic of ongoing work.

**Data Availability**

All data (reference monitor data, RAMP raw signal data, calibrated RAMP data for both training and testing), and codes (in

MATLAB language) to recreate the results discussed here are provided online at https://doi.org/10.5281/zenodo.1302030 (Malings, 2018).

**Acknowledgements**

Funding for this study is provided by the Environmental Protection Agency (Assistance Agreement No. 83628601) and the Heinz Endowment Fund (Grants E2375 and E3145). The authors thank Aja Ellis, Provat K. Saha, and S. Rose Eilenberg for





their assistance with deploying and maintaining the RAMP network and Ellis S. Robinson for assistance with the CMU collocation site.

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





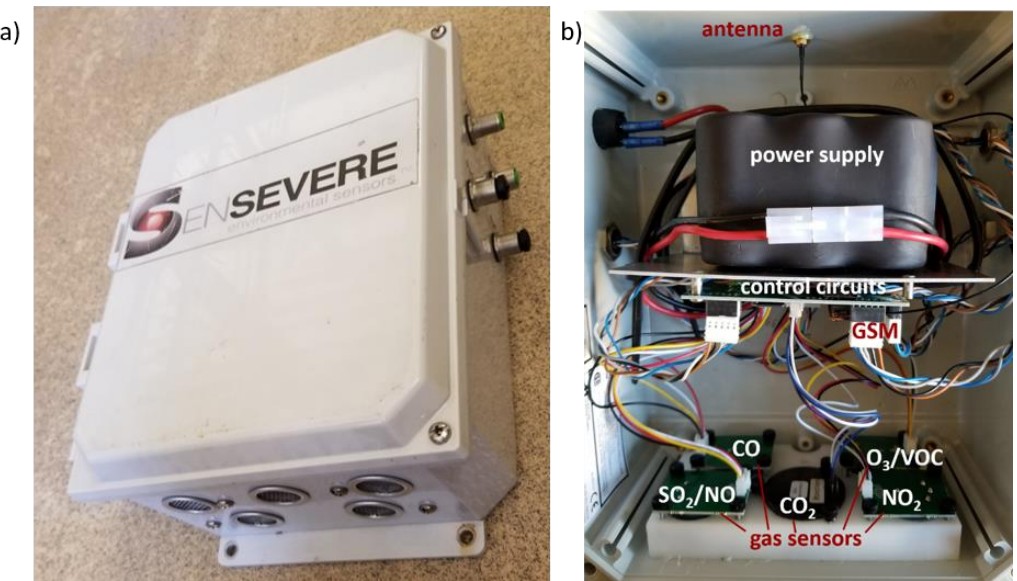

Figure 1: External (a) and internal (b) configuration of the RAMP monitor.





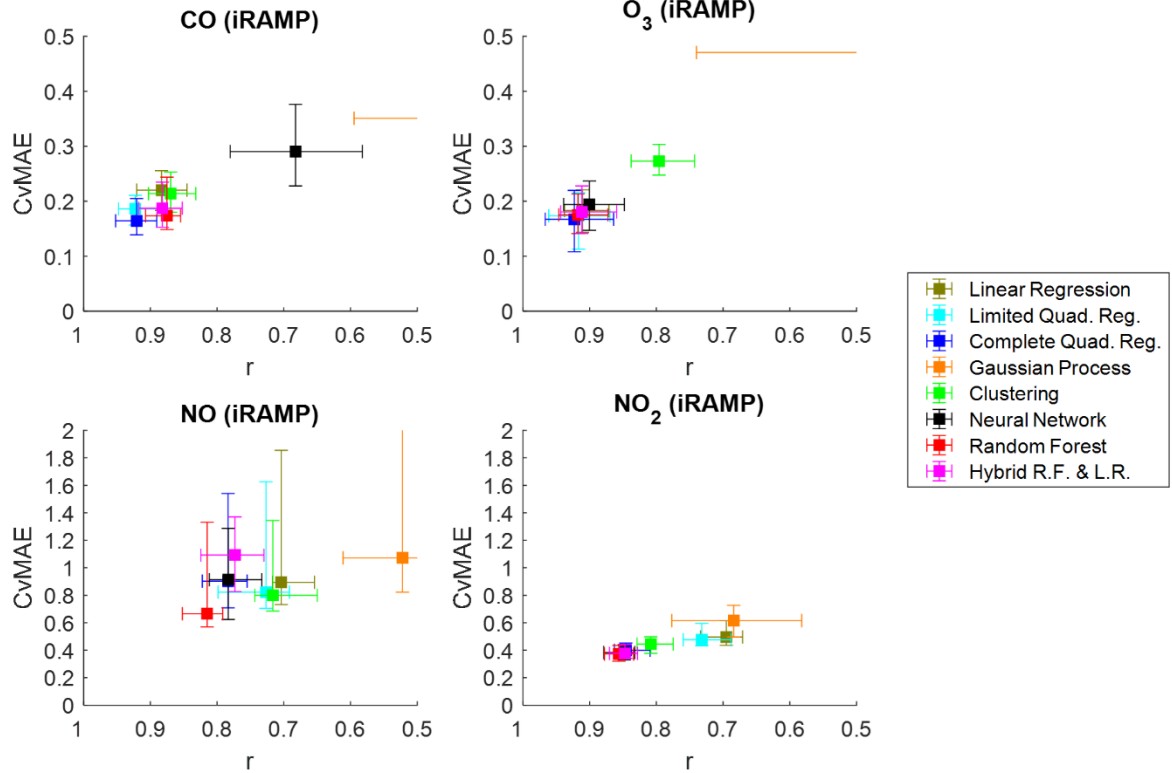

**Figure 2.** Comparative performance of various individualized RAMP calibration models across gases measured by the RAMPs. Models are trained and tested on distinct subsets of collocation data collected at the CMU site during 2017; performance shown is based on the testing data set only. Proximity to the lower-left corner of each figure indicates better performance. Note the differing vertical axis scales.





**Figure 3. Comparative performance of individualized (iRAMP), best individual (bRAMP), and general (gRAMP) model categories across gases measured by the RAMPs. The modelling algorithms used for each gas corresponds to three of the better-performing algorithms identified among the individualized models. Models are trained and tested on distinct subsets of collocation data collected at the CMU site during 2017; performance shown is based on the testing data set only. Proximity to the lower-left corner of each figure indicates better performance.**



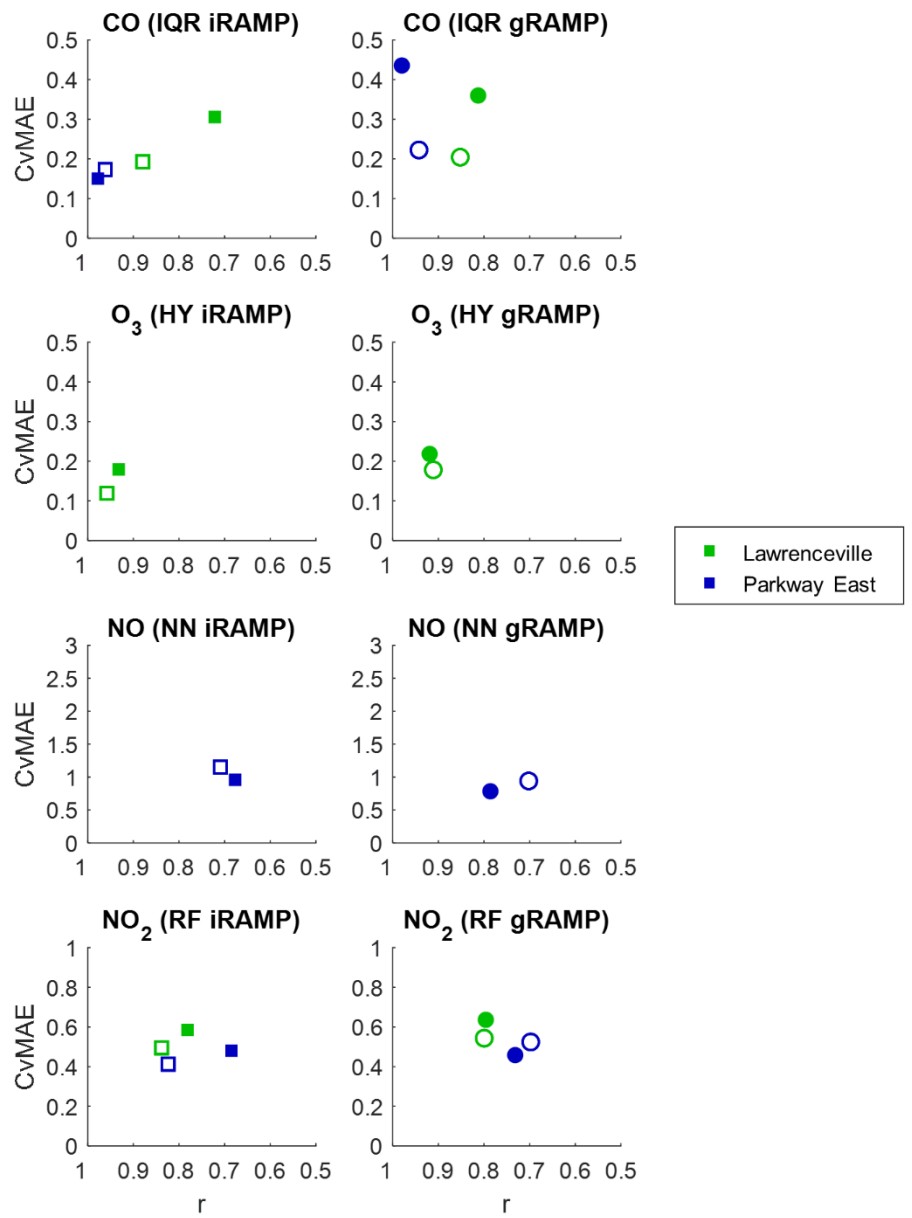

**Figure 4. Comparative performance of individual and general models for RAMPs deployed to ACHD monitoring stations (filled makers), compared to the performance of the same RAMPs at the CMU site (hollow markers). The modelling algorithm used for each gas corresponds to the most consistent algorithm identified among the models depicted in Fig. 3: limited quadratic regression (lQR) for CO, neural network (NN) for NO, random forest (RF) for NO₂, and hybrid linear and random forest (HY) for O₃. Models are trained on data collected at the CMU site during 2017; performance shown for the CMU site (hollow marker) is based on the testing data for the corresponding RAMPs collected at that site. Proximity to the lower-left corner of each figure indicates better performance.**


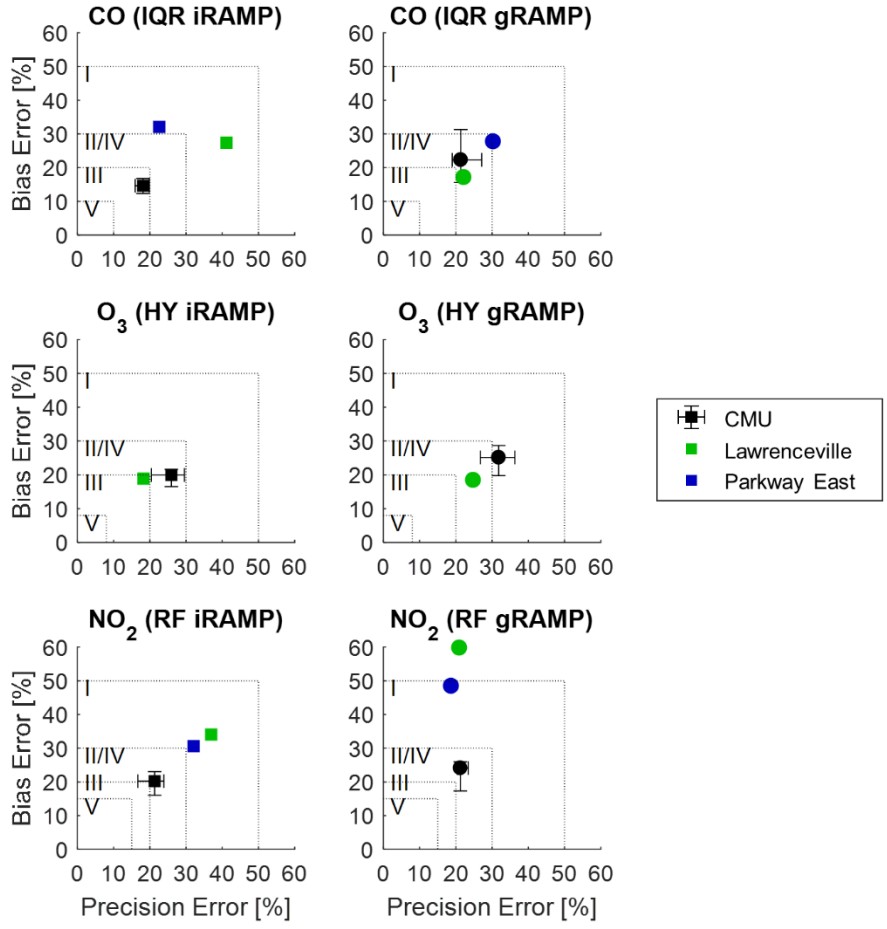

**Figure 5. Comparative performance of individual and general models for RAMPs deployed to ACHD monitoring stations using EPA performance criteria. Dotted lines indicate the outer limits of each performance tier. Performance shown for the CMU site is based on performance across all RAMPs at that site based on testing data only. Proximity to the lower-left corner of each figure indicates better performance.**





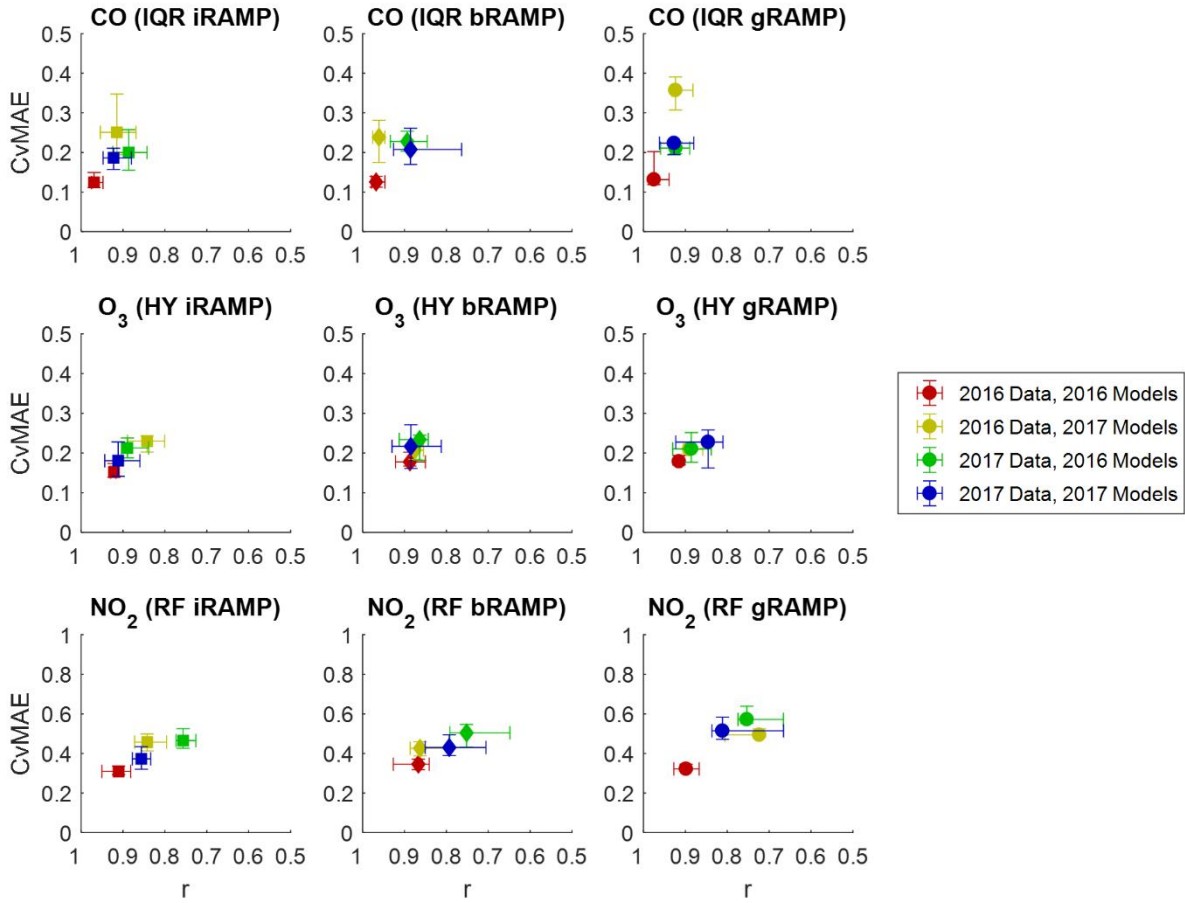

**Figure 6. Comparative performance of individualized random forest hybrid models in 2016 and 2017. Models for 2016 are trained using data collected at the CMU site during 2016 or 2017, and tested on either set as indicated.**





**Figure 7. Tracking the performance of RAMP monitors deployed to ACHD Lawrenceville, ACHD Parkway East, and CMU over time. Statistics are computed for each week. Results shown correspond to those of models trained using data collected at the CMU site during 2017. For CO, the generalized limited quadratic regression model is used; for NO₂, a general random forest model is used; for O₃, the generalized hybrid linear and random forest model is used.**



**Table 1: Assigned lower limits for censoring small measurement values.**

| Quantity | Assigned Lower Limit |
|----------|---------------------|
| CO | 200 ppb |
| $NO_2$ | 10 ppb |
| $O_3$ | 10 ppb |

**Table 2: EPA air quality sensor performance guidelines for various applications. Reproduced from (Williams et al., 2014).**

| Tier | Application | Error Metrics |
|------|-------------|---------------|
| I | Education | < 50% |
| II | Hotspot Identification and Characterization | < 30% |
| III | Supplemental Monitoring | < 20% |
| IV | Personal Exposure Monitoring | < 30% |
| V | Regulatory Monitoring | < 7% for $O_3$ |
| | | < 10% for CO |
| | | < 15% for $NO_2$ |

5    **Table 3: Performance of iRAMP calibration models with respect to EPA air quality sensor performance guidelines as assessed at the CMU site. Entries in the table denote which models meet the corresponding guidelines for each gas (LR = linear regression; lQR = limited quadratic regression; cQR = complete quadratic regression; GP = Gaussian process; CL = clustering; NN = neural network; RF = random forest; HY = hybrid random forest and linear regression).**

| Gas | Tier I | Tier II/IV | Tier III |
|-----|--------|------------|----------|
| CO | NN | LR, GP, CL, HY | lQR, cQR, RF |
| $O_3$ | CL | LR, lQR, cQR, GP, NN, RF, HY | |
| $NO_2$ | GP | LR, lQR, cQR, CL, NN, RF, HY | |





**Table 4: Performance data for iRAMP models at CMU in 2017 (Avg. is the average, SD is the standard deviation). Durations of the training and testing periods are in days. Concentration, MAE, and bias are in units of ppb for all gases except CO$_2$, which uses units of ppm. Models indicates the total number of iRAMP models considered. Slope and r$^2$ are presented for the best-fit-line between the calibrated RAMP measures and those of the regulatory monitor.**

| Gas | Model | Training Period Duration [days] Range | Training Period Concentration [ppb/ppm] Avg. | Training Period Concentration [ppb/ppm] Range | Testing Period Duration [days] Range | Testing Period Concentration [ppb/ppm] Avg. | Testing Period Concentration [ppb/ppm] Range | Models | Slope Avg. | Slope SD | r² Avg. | r² SD | MAE [ppb/ppm] Avg. | MAE [ppb/ppm] SD | Bias [ppb/ppm] Avg. | Bias [ppb/ppm] SD |
|---|---|---|---|---|---|---|---|---|---|---|---|---|---|---|---|---|
| CO | LR | 21 - 28 | 265 | 7 - 3750 | 3 - 75 | 242 | 7 - 3750 | 48 | 1.08 | 0.36 | 0.75 | 0.16 | 60 | 19 | -6 | 18 |
| . | lQR | . | . | . | . | . | . | . | 1.01 | 0.18 | 0.83 | 0.10 | 48 | 11 | -5 | 14 |
| . | cQR | . | . | . | . | . | . | . | 0.96 | 0.16 | 0.85 | 0.09 | 46 | 16 | -3 | 20 |
| . | CL | . | . | . | . | . | . | . | 1.06 | 0.24 | 0.74 | 0.11 | 58 | 17 | 1 | 22 |
| . | NN | . | . | . | . | . | . | . | 1.33 | 1.11 | 0.46 | 0.23 | 84 | 34 | -2 | 34 |
| . | RF | . | . | . | . | . | . | . | 1.19 | 0.30 | 0.78 | 0.09 | 52 | 18 | -2 | 22 |
| . | HY | . | . | . | . | . | . | . | 0.94 | 0.20 | 0.77 | 0.12 | 52 | 15 | 11 | 22 |
| NO | LR | 26 - 28 | 1.98 | 0 - 66 | 4 - 93 | 1.70 | 0 - 66 | 19 | 1.31 | 0.56 | 0.15 | 0.07 | 2.3 | 1.1 | 0.26 | 0.80 |
| . | lQR | . | . | . | . | . | . | . | 1.15 | 0.52 | 0.25 | 0.15 | 2.3 | 1.1 | 0.26 | 0.80 |
| . | cQR | . | . | . | . | . | . | . | 0.97 | 0.36 | 0.36 | 0.14 | 2.1 | 1.0 | 0.35 | 0.82 |
| . | CL | . | . | . | . | . | . | . | 0.67 | 0.33 | 0.18 | 0.10 | 2.2 | 1.2 | 0.08 | 0.80 |
| . | NN | . | . | . | . | . | . | . | 0.90 | 0.35 | 0.30 | 0.13 | 2.0 | 1.0 | 0.09 | 0.55 |
| . | RF | . | . | . | . | . | . | . | 1.14 | 0.42 | 0.41 | 0.12 | 1.9 | 0.9 | 0.14 | 0.67 |
| . | HY | . | . | . | . | . | . | . | 0.65 | 0.37 | 0.32 | 0.14 | 2.3 | 0.8 | 0.76 | 0.66 |
| NO$_2$ | LR | 22 - 28 | 6.52 | 0 - 31 | 4 - 110 | 6.35 | 0 - 32 | 62 | 0.89 | 0.31 | 0.17 | 0.08 | 3.4 | 0.7 | 0.16 | 0.88 |
| . | lQR | . | . | . | . | . | . | . | 0.79 | 0.23 | 0.21 | 0.09 | 3.3 | 0.6 | 0.18 | 0.93 |
| . | cQR | . | . | . | . | . | . | . | 0.85 | 0.15 | 0.47 | 0.11 | 2.6 | 0.5 | 0.07 | 0.73 |
| . | CL | . | . | . | . | . | . | . | 0.77 | 0.17 | 0.37 | 0.12 | 2.9 | 0.5 | 0.27 | 0.70 |
| . | NN | . | . | . | . | . | . | . | 0.93 | 0.16 | 0.49 | 0.12 | 2.6 | 0.5 | 0.07 | 0.59 |
| . | RF | . | . | . | . | . | . | . | 0.97 | 0.14 | 0.50 | 0.10 | 2.5 | 0.5 | 0.29 | 0.64 |
| . | HY | . | . | . | . | . | . | . | 0.83 | 0.13 | 0.48 | 0.10 | 2.6 | 0.4 | 0.51 | 0.63 |
| O$_3$ | LR | 21 - 28 | 25.6 | 1 - 128 | 2 - 76 | 28.8 | 1 - 128 | 44 | 0.98 | 0.06 | 0.80 | 0.12 | 5.1 | 1.7 | -0.05 | 1.6 |
| . | lQR | . | . | . | . | . | . | . | 0.96 | 0.05 | 0.83 | 0.11 | 4.6 | 1.7 | 0.12 | 1.4 |
| . | cQR | . | . | . | . | . | . | . | 0.93 | 0.07 | 0.82 | 0.12 | 4.6 | 1.7 | -0.08 | 1.1 |
| . | CL | . | . | . | . | . | . | . | 0.89 | 0.11 | 0.62 | 0.11 | 7.3 | 1.3 | -0.47 | 2.4 |
| . | NN | . | . | . | . | . | . | . | 0.98 | 0.21 | 0.73 | 0.26 | 5.8 | 2.8 | 0.09 | 1.3 |
| . | RF | . | . | . | . | . | . | . | 1.00 | 0.06 | 0.83 | 0.08 | 4.7 | 1.3 | -0.09 | 1.5 |
| . | HY | . | . | . | . | . | . | . | 0.93 | 0.06 | 0.81 | 0.09 | 4.9 | 1.3 | 0.40 | 1.5 |
| CO$_2$ | LR | 21 - 28 | 434 | 365 - 567 | 2 - 50 | 425 | 365 - 601 | 38 | 0.74 | 0.23 | 0.21 | 0.09 | 24 | 5 | 0.6 | 6.1 |
| . | lQR | . | . | . | . | . | . | . | 0.62 | 0.22 | 0.23 | 0.12 | 25 | 5 | 1.0 | 8.1 |
| . | cQR | . | . | . | . | . | . | . | 0.74 | 0.20 | 0.47 | 0.16 | 18 | 3 | -1.0 | 4.8 |
| . | CL | . | . | . | . | . | . | . | 0.76 | 0.13 | 0.43 | 0.13 | 20 | 3 | 1.4 | 4.4 |
| . | NN | . | . | . | . | . | . | . | 0.47 | 1.53 | 0.28 | 0.25 | 23 | 7 | -2.1 | 6.5 |
| . | RF | . | . | . | . | . | . | . | 0.93 | 0.18 | 0.56 | 0.12 | 17 | 2 | 0.6 | 4.3 |
| . | HY | . | . | . | . | . | . | . | 0.79 | 0.26 | 0.53 | 0.15 | 19 | 4 | 3.2 | 6.0 |

