# Peer review of "Development of a General Calibration Model and Long-Term Performance Evaluation of Low-Cost Sensors for Air Pollutant Gas Monitoring"

_Atmospheric Measurement Techniques, 2018_

## Referee Comment (RC1) · Anonymous Referee #1 · 24 Oct 2018

This paper examines various approaches for calibrating low-cost air quality (AQ) sensors, with the goal of making recommendations for effective calibration strategies, especially over relatively long timescales (months to years). This is an important and timely topic in atmospheric chemistry, and is certainly will be of interest to the readership of AMT. The key result, that a "generalized calibration model" - in which a number of sensors are calibrated via collocation at EPA-grade AQ monitoring sites, and the average calibration be used for all sensors – provides adequate accuracy for many applications, is certainly a useful and important one. However, a weakness of this paper is that the analysis approaches taken are not always clearly described (or well-justified) in the manuscript, so it is not always obvious how general the conclusions are. In par-

ticular, the calibration approach over longer terms is not well-described, and appears to involve an test/training approach that is different than what would be used under most conditions. Thus the general validity of the recommendations (that new models should be developed every year) is unclear. These issues, described below, should be addressed before this paper is published in AMT.

(format is "pageNumber_lineNumber")

The way one would calibrate sensors under standard deployment conditions is to collocate at an EPA station for some period of time (or to calibrate in the lab), then deploy the sensor to some other location of interest to make new measurements (possibly returning the sensor to the collocation spot later on for re-calibration). But this doesn't appear to be the approach taken here, where the long-term data (Figure 6 and 7, and accompanying text, p. 14-15) seems to have training/test data taken throughout a given year. (Though this isn't well-described in the manuscript – what were the training and test times? How were these chosen?) If the test data is indeed taken throughout the year, this isn't really a realistic calibration approach, so it is unclear to me how the authors can make recommendations about how or how often calibrations should be done (1_20, 14_29).

Similarly, from Table 4, it appears the training and test sets cover nearly identical ranges in pollutant concentrations – to within 1 ppb for the four gases (CO, NO, NO, O3). How is this possible? On implication of these identical ranges is that the performance of the hybrid approach (discussed in sections 2.3.5) cannot really be distinguished from the non-hybrid approaches. This is mentioned near the very end of the paper (17_17), but really should be discussed sooner, and the hybrid and RF-only models probably should not be discussed as separate approaches. If they are, the number of "crossings" (switches from RF to LR, fraction time evaluated by RF vs time evaluated by LR) should be discussed.

4_24-29: The gRAMP approach involves selecting a subset of the sensors for calibration and seeing how the others do with this calibration. However very little information was given on which sensors were used/withheld. Presumably these were sampled randomly (via a k-fold cross-validation, etc), to make sure the selection of sensors in the training set did not bias results?

7_22 (also 2-26): The authors describe the work by Hagan et al. as a clustering approach, but this is incorrect – the authors may be confusing k-nearest-neighbors (kNN, used by Hagan) with k-means-clustering (used in this work). kNN is not a clustering method; clustering in k-means-clustering is computationally much less intensive than storing and comparing to every input-output pair (as is done in kNN), but it can also lead to a dramatic degradation of the quality of the training dataset. Thus, the present k-means-clustering results cannot be compared to the approach of Hagan et al.

Overall: the authors may want to reconsider their terminology, given they are trying to make general recommendations for sensor use, including use of non-RAMP AQ sensors (this is the focus of section 4.1). I would recommend using terms to describe the models that are more general and non-sensor-specific than iRAMP, gRAMP, etc.

Minor comments

- 2_20-22: The wording here should probably be softened somewhat; it is challenging (but not impossible) to access all relevant atmospheric concentrations in the lab.

- 3_21: Small typo: the company name is Alphasense, not AlphaSense.

- 4_24: the gRAMP approach has some similarities to the averaging approach taken by Smith et al. (Faraday Discuss. 2017, 200, 621-637); while there are differences in these two techniques, this previous work should certainly be acknowledged here.

- 5_6-10: how were these cutoffs (15 minutes, 21 days) chosen? It's stated the 15 minute averaging was chosen to reduce noise, but results from other time intervals (1 min, 5 min, 1 hour, etc) are not presented. Are the data so noisy that such averaging is necessary? (Or is this just minute-by-minute variability?)

[Figure]

- Figures 2-3: What do the error bars refer to here - the spread among individual sensors? If possible, it might be more useful to show the data from each individual sensor here.

- 7_30-8_1: since this issue is important to all nonparametric models, as the authors state, this point should be made earlier, not just in the section on k-means-clustering.

- 8_30-32: this sentence implies that the hybrid approach was developed by Zimmerman et al. and used by Hagan et al. My understanding from those two papers (and from the timing of the original AMTD submissions) is that Hagan implemented it, and it was mentioned as a potential approach by Zimmerman.

- 9_13 (section 2.4): this is a very useful section, but it should be highlighted that these metrics are used on the test/validation data only.

- p10-14: Here there is a lot of text describing the individual figures. All this detailed information was rather hard to follow, and hard to glean what the major results were; a "bigger-picture" discussion of what the figures tell us might be helpful.

- Figure 5: how many sensors are we talking about here? Were all of them moved?

- 14_10-12: I don't follow this sentence. If the models are trained and tested on data from both years, how can a change in model performance indicate a change in the models? Do the authors mean a change in the sensors themselves (as discussed in the next paragraph)?

- 14_24: If the sensor is degrading, its output signal will probably be lower for a given amount of pollutant. Is this observed? If not, what evidence is there for degradation, other than a change to the calibration?

- Additionally, in 14_28: I think the problem is not that the electrode material (typically some metal) is "used up" but rather that the electrolyte concentration changes over time, by either evaporation or leaking.
- Table 4: this table is very useful, but a bit hard to follow in its current form. Some suggestions/questions: - since it's not relevant to the text, maybe remove CO2, avoiding the ppb/ppm problem - the concentrations (as measured by FEM/FRM monitors) are in the "LR" row, which might suggest they are relate to the linear regression. Maybe move them to the header of each pollutant, to separate the calibration technique used from the data - the column title "Models" isn't clear - a CO concentration of 7ppb is unusually (maybe impossibly) low – remote regions generally have levels of ∼100 ppb. It might be worth checking that dataset. - T and RH ranges should be included.

16_11-14: This statement is based only on comparisons of sensors run under different conditions at different times and places. Comparisons like this can only really be made when different sensors are studying the same airmass.

- Citations: twice (Hagan et al., Sadighi et al.) the AMTD citation is used rather than the AMT one.

- SI: making the data publicly available is a really excellent feature of this paper, and a good template for other sensor papers. However the file is almost 14GB! It might make more sense to provide just the raw data, and the scripts used; most users will want the data only. Those that want to examine the model output can run the scripts themselves.

---

## Referee Comment (RC2) · Anonymous Referee #2 · 5 Nov 2018

General comments: In this study, authors compared different models used to adjust data from low-cost sensors and compared model application in different sites within the Pittsburgh, Pennsylvania region of the USA. These comparisons demonstrate that at the sites different model-based calibration techniques show differing abilities to accurately produce concentration data for the various sensors employed. However, there existing two main problems: 1 for calibration period and deployment, it is unclear which based on the text, the calibration data also used for testing or vice versa, more details about the deployment should be illustrated. 2 Evaluation validity: As listed in Table 4, the average pollutant concentration is relatively small for NO and NO2 both in training and testing period, which about 1.7 and 6.4 ppb level, how did the investigators ensure

sensor evaluation validity in such low-level situation, while EC sensor has a lower detection limit at about this level? And for NO, the MAE of all models are even large than average concentration.

Specific Comments: 1. Page 1, Line 24: Authors use "These results will help guide future efforts in the calibration and use of low-cost sensor systems worldwide". Perhaps the approaches merit many further trials in the widely differing pollutant and meteorological conditions It is unclear. It is unclear how the study demonstrates protocols that apply worldwide?

2. Page 2, line 16: "These sensors tend to have lower signal-to-noise ratios than regulatory-grade instruments…" is a fairly unusual and non-specific way to say they are not as precise or sensitive as conventional air monitors.

3. Page 2, Line 20: Authors mentioned that a nonlinear interaction in the reference (Jiao et al., 2016), however, this paper didn't mention this kind of nonlinear interaction, but cited Gao's paper (Gao et al., 2015) which also didn't mention this, rather it is a nonlinear response for PM sensor.

4. Page 2, Line 25: Same problem for reference (Cross et al., 2017) as comment 3.

5. Page 3, line 29: It appears that on-campus field calibrations were conducted during a brief period in summer and early fall. Where winter time calibrations performed as well? If not, how might this impact the application of various results of modelling over winter time conditions?

6. At the end Section 2. which describes monitoring methods there is a general reference to the Zimmerman 2018 paper to provide the reader information on protocols. Key factors should be pulled into the current text or figures. Specifically, on Line 30—what were the specific models of regulatory monitors used and how were they located/protected from environmental factors? What methods were employed to measure speciated VOCs mentioned? What is meant by the term "or BTEX"? This does

not appear to be covered in the 2018 Zimmerman paper and it does not seem that VOCs are part of this study—perhaps this needs to be removed from the current text. Further, SO2 is included in the list of pollutants that appear to have been included in this study. However, no data for SO2 appears. Further, several of the models include factors for SO2. Was SO2 measured or employ in this study?

7. Page 4, Line 5: The collocation in ACHD points seems to span the whole year of 2017 in each week at Lawrenceville as shows in Figure 7, while in Table 4 the maximum days of testing period duration only ranged from 75 to 110 days for CO, O3 and NO2. The authors should give more explanation about the deployment? It is unclear and could not be found in the cited reference (Zimmerman et al., 2018). Also, as mentioned in Page 5, Line 9, 80% of collocation is about 28 days less than 1 month, what are other days while RAMP collocation such long time and what is the criteria for period selection?

8. Page 4, Line 10: Authors mentioned ACHD Parkway East has high levels of NO and NO2 for maximum at 100ppb and 40ppb, this period also included in testing period, but in Table 4, for whole testing period these two maximum concentrations are 66 and 32.

9. Page 4, Line 22: The best model was selected based on performance of individual models, what is the criteria for this?

10. Page 9, Line 4: The random forest and hybrid model used all sensor data for training CO, and combined linear model which only used CO, why not introduce other sensors in this procedure?

11. Page 10, Line 8: As mentioned in the text "measurements where the corresponding true value is below an assigned lower limit are removed from the measurement set to be evaluated", while for NO2(10ppb) and O3 (10ppb) which listed in Table 1, the training period or testing period concentration range still start from 0 or 1ppb, does any filter about lower limit come into force in the training or evaluation? The authors should discuss the impact of removal of data below specified minimums as this would

[Figure]

appear to skew the actual data and do so in a differential basis depending on ambient conditions.

12. Page 16, line 11: "The fact that for most gases a variety of calibration approaches show similar and (for typical uses cases, acceptable) performance may reflect better underlying performance from the RAMP monitor, as similar studies for other low-cost sensor packages showed a wider variability in performance between calibration approaches (see e.g. the summary provided by Zimmerman et al., 2018)." This statement appears to indicate that the RAMP monitors somehow performed better than other similar sensor based monitors. However, there is no rationale for this statement based on the actual monitor package, component details provided or information reported by others. It appears to simply be a diffusion based sensor deployment of Alfasense electrochemical cells and an NDIR CO2 sensor. What other factors might influence system performance?

13. Page 25, Figure 4: The performance of CMU site for each sensor is marked in hollow marker, while there are two hollow markers in CO and NO2, which is unclear.

14. Page 30, Table 4: In the evaluation of testing period, R2 was used, however, in previous figure and text, such as Figure 2-4, Pearson linear correlation coefficient r was used, what is the standard for selection between these two parameters?

15. Page 30, Table 4: The title of the paper is about a general calibration model (gRAMP) and recommended in the conclusion, while it was not compared with iRAMP models in Table 4.

This paper focuses on field calibration methods and model application to produce adjusted pollutant concentration data from sensor based monitors. However, discussions and recommendations are only made regarding field calibration approaches. It would also appear that the data agreement between sensors and regulatory monitoring might be improved by other means. For example, conventional air monitoring methods as applied to regulatory monitoring efforts always include periodic zero and span challenges.

Improved circuitry or air conditioning might aslo improve monitor performance. How might the inclusion of such methods improve the quality of data produced by sensor based systems?

---

## Author Comment (AC1) · 3 Dec 2018

We would like to thank the reviewer for their constructive comments. We have tried to address these comments in the attached response document and in the manuscript. Reviewer comments are reproduced in black, our responses are in blue.

**General Comments**

This paper examines various approaches for calibrating low-cost air quality (AQ) sensors, with the goal of making

- 5 recommendations for effective calibration strategies, especially over relatively long timescales (months to years). This is an important and timely topic in atmospheric chemistry, and is certainly will be of interest to the readership of AMT. The key result, that a "generalized calibration model" in which a number of sensors are calibrated via collocation at EPA-grade AQ monitoring sites, and the average calibration be used for all sensors provides adequate accuracy for many applications, is certainly a useful and important one. However, a weakness of this paper is that the analysis approaches taken are not always
- 10 clearly described (or well-justified) in the manuscript, so it is not always obvious how general the conclusions are. In particular, the calibration approach over longer terms is not well-described, and appears to involve an test/training approach that is different than what would be used under most conditions. Thus the general validity of the recommendations (that new models should be developed every year) is unclear. These issues, described below, should be addressed before this paper is published in AMT. (format is "pageNumber\_lineNumber")
- 15 The way one would calibrate sensors under standard deployment conditions is to collocate at an EPA station for some period of time (or to calibrate in the lab), then deploy the sensor to some other location of interest to make new measurements (possibly returning the sensor to the collocation spot later on for re-calibration). But this doesn't appear to be the approach taken here, where the long-term data (Figure 6 and 7, and accompanying text, p. 14-15) seems to have training/test data taken throughout a given year. (Though this isn't well-described in the manuscript – what were the training and test times? How were these
- 20 chosen?) If the test data is indeed taken throughout the year, this isn't really a realistic calibration approach, so it is unclear to me how the authors can make recommendations about how or how often calibrations should be done (1\_20, 14\_29). For the results discussed, training of calibration models takes place at specific times and locations, depending on the case being discussed. In the results relating to Figures 2 and 3, specific subsets of the data collected by the RAMPs when they are deployed at the CMU site in 2017, amounting to a maximum total of 28 days of training data per RAMP, are used to develop the models;
- 25 the performance of the models is evaluated on whatever other data is available from this site which was not used in the training. This has been clarified in the text (5 26-6 7):

"From the collocation data, eight equally sized, equally spaced time intervals are selected to serve as training data for the calibration models. The amount of training data is selected to be either 80% of the collocation data or four weeks of data (corresponding to 2688 15-minute-averaged data points), whichever is smaller. The minimum amount of training data is 21 days; if less than this is available, no iRAMP model is trained for this RAMP, and thus no iRAMP model performance can be assessed for it (although bRAMP and gRAMP models trained on other RAMPs are still applied to this RAMP for testing). Training data for gRAMP models are obtained in the same way, although in that case it is the data for the virtual "typical RAMP" which are divided, rather than data for individual RAMPs. Any remaining data from the collocation period are left aside as a separate testing set, on which the performance of the trained models is evaluated. Note that due to differences in which RAMPs and/or regulatory-grade instruments were operating at a given time, training and testing periods are not necessarily the same for all RAMPs and gases; for example, a certain time may be part of the training period for the CO model for one RAMP, and be part of the testing period for the O3 model of another RAMP. However, the training and testing periods for a given RAMP and gas are always distinct. The division of data collected at the CMU site in 2017 into training and testing periods is illustrated in the supplemental information (Figures S6-S10). The division of data collected at the CMU site in 2016 is carried out in a similar manner. The choice of averaging period, of minimum and maximum training times, and the method for dividing between training and testing periods are motivated by previous work with the RAMP monitors (Zimmerman et al., 2018). "

10 In addition, a series of figures has been added to the supplemental information, detailing for each RAMP and each gas sensor which data collected at the CMU site are used for training and which are used for testing.

For Figures 4, 5, and 7, performance at the deployment sites (Lawrenceville and Parkway East) are assessed using models already developed at the CMU site (the performance of which at that site are represented as hollow markers in Figure 4, black markers in Figure 5, and black lines in Figure 7); therefore, all data collected at these deployment sites are in effect treated as

- 15 "testing data". In other words, no site-specific training is done for these deployed sensors. We believe this is similar to the typical use case described by the reviewer, namely that a sensor is first collocated at a reference station (in this case, not an EPA station, but rather a similar station set up at the CMU campus) to allow for model calibration, and then deployed to another site to collect data. The fact that these other sites were EPA stations allowed us to have access to "ground truth" data for assessing the performance of our calibrations, but these data were not used to develop new calibrations for the deployed
- 20 sensors, as this would not represent a realistic use case in general. This has been further clarified in the text (6\_7-9): "All data collected at sites other than the CMU site (i.e. the Lawrenceville or Parkway East sites) are reserved for testing; no training of calibration models is done using data collected at these other sites, and so they represent a true test of the performance of the models at an "unseen" location."

For Figure 6, models trained in 2016 or 2017 are trained using only a portion of the data collected in that year, and their
performance is evaluated on another distinct subset of data collected in that year. For 2017, these training and testing subsets are exactly the same as those used for the results of Figures 2 and 3. For 2016, the sets are different, but are determined using the same method as was used for the 2017 data. This has been further clarified in the text (6\_3-5):

"The division of data collected at the CMU site in 2017 into training and testing periods is illustrated in the supplemental information (Figures S6-S10). The division of data collected at the CMU site in 2016 is carried out in a similar manner."

30

5

And (14\_8-11):

"Training and testing data for 2017 represent the same training and testing periods as used for previous results. For 2016, training and testing data are divided using the same procedure as was applied for 2017 data, as discussed in Sect. 2.3. For example, the results for "2016 Data, 2017 Models" represent the performance of models calibrated

using the training data subset of the 2017 CMU site data when applied to the testing data subset of the 2016 CMU site data. "

Similarly, from Table 4, it appears the training and test sets cover nearly identical ranges in pollutant concentrations – to within 1 ppb for the four gases (CO, NO, NO, O3). How is this possible?

- 5 Because Table 4 refers to training and testing data sets used for the iRAMP models, each RAMP has its own training and testing data sets. However, because not all RAMPs were present and operating at the CMU site at the same times, the time period encompassing the training set for one RAMP may be part of the testing set for a different RAMP, and vice versa. Thus, it is quite common for high and low concentrations to show up in the training sets of some RAMPs and the testing sets of others, and thus be reported in this table as being part of both the training and testing set ranges. We have attempted to clarify
- 10 this by instead reporting ranges of high, average, and low concentrations for the training and testing sets in this table. Furthermore, in the supplemental information, we have included several figures describing the distribution of measurements in the training and testing sets for each RAMP. Finally, we have divided this table into two tables, one presenting the concentration ranges and the other depicting the performance information.

One implication of these identical ranges is that the performance of the hybrid approach (discussed in sections 2.3.5) cannot really be distinguished from the non-hybrid approaches. This is mentioned near the very end of the paper (17\_17), but really should be discussed sooner, and the hybrid and RF-only models probably should not be discussed as separate approaches. If they are, the number of "crossings" (switches from RF to LR, fraction time evaluated by RF vs time evaluated by LR) should be discussed.

As the reviewer suggests, due to the large degree of overlap between these models, we have removed the separate discussion of results from both model types, and instead focus on the hybrid approach only.

20

4\_24-29: The gRAMP approach involves selecting a subset of the sensors for calibration and seeing how the others do with this calibration. However very little information was given on which sensors were used/withheld. Presumably these were sampled randomly (via a k-fold cross-validation, etc), to make sure the selection of sensors in the training set did not bias results?

- 25 Selection of the training and testing sets for the gRAMP model was done randomly; RAMPs were included in the training set with a probability of 80%. Following this, a few manual adjustments were to the selected sets were made, such that RAMPs which were to be deployed to the Lawrenceville and Parkway East sites were not included in the training data set. However, this selection was made only once, and not resampled. This has been clarified in the manuscript (5\_11-15):
- "RAMPs were divided into training and testing sets for the gRAMP models randomly, with the caveat that the two
   RAMPs deployed to the Lawrenceville and Parkway East sites were required to be part of the testing set. Data from about three quarters of the RAMP monitors (53 out of 68) were used for developing the general calibration models (although not all of these monitors were active at the same time). Data from the remaining 15 RAMP monitors were used for testing, ensuring that the testing data are completely distinct from the training data."

7\_22 (also 2-26): The authors describe the work by Hagan et al. as a clustering approach, but this is incorrect – the authors may be confusing k-nearest-neighbors (kNN, used by Hagan) with k-means-clustering (used in this work). kNN is not a clustering method; clustering in k-means-clustering is computationally much less intensive than storing and comparing to every input-output pair (as is done in kNN), but it can also lead to a dramatic degradation of the quality of the training dataset. Thus, the present k-means-clustering results cannot be compared to the approach of Hagan et al.

As described by the reviewer, the clustering approach presented in this paper is a variation of the k-nearest-neighbors approach in which clustering is used to reduce the number of stored input-output pairs; this improved the computational efficiency of the approach at the expense of possibly lower performance. This distinction has been made clear in the text (8\_14-18):

"In a traditional k-nearest-neighbors approach, such as that used in previous work (Hagan et al., 2018), every input-

10 output pair from the training data is stored for comparison to new inputs. Although this provides the best possible estimation performance via this approach, storing these data and performing these comparisons are computation- and memory-intensive. Therefore, in this work, the input data are first clustered, i.e., grouped by proximity of the input data."

Overall: the authors may want to reconsider their terminology, given they are trying to make general recommendations for sensor use, including use of non-RAMP AQ sensors (this is the focus of section 4.1). I would recommend using terms to describe the models that are more general and non-sensor-specific than iRAMP, gRAMP, etc.

In the revised manuscript, we make use of the "iRAMP", "bRAMP", and "gRAMP" acronyms to describe the models when they are specifically applied to the data collected from the RAMP sensors. Otherwise, when discussing these approaches generally and drawing conclusions, we make use of the less specific terminology, e.g., "generalized" or "individualized"

20 models. This has been explained in the text  $(5_19-21)$ :

"Finally, note that, for brevity, we will refer to iRAMP, bRAMP, or gRAMP model variants when discussing specific results; however, when drawing general conclusions about low-cost electrochemical gas sensor calibration methods, we will use less RAMP-specific terms (such as "generalized models")."

**Minor comments**

5

25 - 2\_20-22: The wording here should probably be softened somewhat; it is challenging (but not impossible) to access all relevant atmospheric concentrations in the lab.

This has been corrected (2\_21-23):

"Due to the variety of interactions and atmospheric conditions which can affect sensor performance, covering the range of conditions to which the sensor will be exposed using laboratory calibrations is difficult."

**30 - 3\_21: Small typo: the company name is Alphasense, not AlphaSense.**

Thank you for pointing this out; it has been corrected.

- 4\_24: the gRAMP approach has some similarities to the averaging approach taken by Smith et al. (Faraday Discuss. 2017, 200, 621-637); while there are differences in these two techniques, this previous work should certainly be acknowledged here.

Thank you for bringing this to our attention. The motivating ideas are indeed similar, but in our case we use the median of a sensor ensemble to generate data for calibration, and then apply this calibration to the outputs of individual sensors to evaluate performance. This discussion has been added to the manuscript (5\_7-10):

"The motivation for the use of gRAMP models is similar to that of Smith el at. (2017); however, while in that work

it is recommended that the median from a set of duplicate low-cost sensors be used to improve performance, in this work we use that method to develop the gRAMP calibration model, but then apply this calibration to the outputs of individual sensors rather than to the median of a group of sensors."

- 5\_6-10: how were these cutoffs (15 minutes, 21 days) chosen? It's stated the 15 minute averaging was chosen to reduce noise, but results from other time intervals (1 min, 5 min, 1 hour, etc) are not presented. Are the data so noisy that such averaging

10 is necessary? (Or is this just minute-by-minute variability?)

5

30

The choice of 15 minutes as an averaging period, the upper and lower limits on the amount of training data, and the method by which these data are divided are motivated by previous work with the RAMP sensors (Zimmerman et al., AMT, 2018, 11, 291-313). Recently we have examined the performance of the calibration models when applied to raw RAMP data using different averaging periods (ranging from 1 minute to 1 day). These results are included in the supplemental information, and

15 indicate that performance is relatively stable for averaging periods below 1 hour (14\_31-15\_5):
"Additionally, calibration model performance was assessed as a function of averaging time. Note that the calibration models discussed in this paper are developed using RAMP data averaged over 15-minute intervals, as discussed in Section 2.3. However, these models may be applied to raw RAMP signals averaged over longer or shorter time periods. Furthermore, the calibrated data can also be averaged over different periods. To investigate the effects of averaging time on calibration model performance, we assess the performance of RAMPs calibrated with gRAMP models for CO, O3, and CO2 at the CMU site in 2017, with averaging performed either before or after the calibration. Results are provided in the supplemental materials (Figures S4 and S5). Overall, we find little variation in calibration model performance with respect to averaging periods between 1 minute and 1 hour."

- Figures 2-3: What do the error bars refer to here - the spread among individual sensors? If possible, it might be more useful

25 to show the data from each individual sensor here.

Error bars indicate the interquartile range in performance across RAMPs. We originally had a version of this figure which presented each performance of each RAMP with a single point; this proved to be very difficult to interpret, which is why we chose to present the results in this way.

- 7\_30-8\_1: since this issue is important to all nonparametric models, as the authors state, this point should be made earlier, not just in the section on k-means-clustering.

This discussion has been moved to the beginning of the section on calibration models (4\_25-30):

"A common difficulty of non-parametric methods is generalizing beyond the training data set. For example, if no high concentrations are observed during the collocation period, then the resulting trained nonparametric model will be unable to estimate such high concentrations if it is exposed to these during deployment. This is of potential concern

for air quality applications, as the detection of high concentrations is an important consideration. Parametric models avoid this difficulty, but at the cost of lower flexibility in the types of input-output relationships they can capture."

- 8\_30-32: this sentence implies that the hybrid approach was developed by Zimmerman et al. and used by Hagan et al. My understanding from those two papers (and from the timing of the original AMTD submissions) is that Hagan implemented it,

**5 and it was mentioned as a potential approach by Zimmerman.**

Correct. This has been re-worded to clarify that (9\_24-26):

"The use of this approach for RAMP data was suggested by Zimmerman et al. (2018). Furthermore, it is similar to the approach of Hagan et al. (2018), who hybridize nearest neighbor and linear regression models."

**- 9\_13 (section 2.4): this is a very useful section, but it should be highlighted that these metrics are used on the test/validation**

**10 data only.**

This has been explicitly stated at the beginning of the section  $(10_7-9)$ :

"It should be noted that the metrics presented here are applied only for testing data, i.e., data which were not used to build the calibration models. Model performance on the training data is expected to be higher, and thus less representative of the true capability of the model."

- p10-14: Here there is a lot of text describing the individual figures. All this detailed information was rather hard to follow, and hard to glean what the major results were; a "bigger-picture" discussion of what the figures tell us might be helpful.
 Much of these detailed results have been omitted, and instead more emphasis has been placed on the conclusions drawn from these results.

**- Figure 5: how many sensors are we talking about here? Were all of them moved?**

20 For the results of Figures 4, 5, and 7, only one sensor is present at each of the deployment sites (i.e. one sensor at Lawrenceville and one sensor at Parkway East). This has been clarified in the text (13\_2-3):

"Figure 4 depicts the performance of calibration models for two RAMP monitors deployed at two EPA monitoring stations operated by the ACHD (one monitor is deployed to each station)."

- 14\_10-12: I don't follow this sentence. If the models are trained and tested on data from both years, how can a change in

25 model performance indicate a change in the models? Do the authors mean a change in the sensors themselves (as discussed in the next paragraph)?

Models are trained on a subset of data collected in one year, and then tested either on a distinct subset of the data from that year or on a testing data subset from the other year. This has been clarified in the text (14\_8-11):

30

"Training and testing data for 2017 represent the same training and testing periods as used for previous results. For 2016, training and testing data are divided using the same procedure as was applied for 2017 data, as discussed in Sect. 2.3. For example, the results for "2016 Data, 2017 Models" represent the performance of models calibrated using the training data subset of the 2017 CMU site data when applied to the testing data subset of the 2016 CMU site data."

Changes in performance on data collected in the same year are due only to the differences in the models; changes in performance on data collected in different years using the same model are only due to differences in the sensor responses.

- 14 24: If the sensor is degrading, its output signal will probably be lower for a given amount of pollutant. Is this observed? If not, what evidence is there for degradation, other than a change to the calibration?

This has been observed in some data recently collected from these sensors (14 20-24): 5

> "Thus, a model calibrated on the response characteristics of the sensors in one year will not necessarily perform as well using data collected by the same sensors in a different year. This degradation has also been directly observed, as the raw responses of "old" sensors deployed with the RAMPs since 2016 were compared to those of "new" sensors recently purchased in 2018; in some cases, responses of "old" sensors were about half the amplitude of those of "new"

10

15

sensors exposed to the same conditions."

- Additionally, in 14 28: I think the problem is not that the electrode material (typically some metal) is "used up" but rather that the electrolyte concentration changes over time, by either evaporation or leaking.

Thank you for pointing this out; it has been corrected (14 24-25):

"This is consistent with the operation of the electrochemical sensors, where the electrolyte concentration changes over time as part of the normal functioning of the sensor."

- Table 4: this table is very useful, but a bit hard to follow in its current form. Some suggestions/questions: - since it's not relevant to the text, maybe remove CO2, avoiding the ppb/ppm problem - the concentrations (as measured by FEM/FRM monitors) are in the "LR" row, which might suggest they are relate to the linear regression. Maybe move them to the header of each pollutant, to separate the calibration technique used from the data - the column title "Models" isn't clear - a CO

concentration of 7ppb is unusually (maybe impossibly) low – remote regions generally have levels of 100 ppb. It might 20 be worth checking that dataset. - T and RH ranges should be included.

Thank you for these suggestions. This table has been divided in two, with the first table displaying concentration information (as well as ranges for T and RH) and the second showing the performance of iRAMP models. Information for CO2 has been moved to the supplemental information. Finally, it appears the report of 7 ppb was a typo, it was meant to be 57 ppb.

16 11-14: This statement is based only on comparisons of sensors run under different conditions at different times and places. 25 Comparisons like this can only really be made when different sensors are studying the same airmass.

This is based on reported differences between calibration model performances in the literature. The signal conditioning employed in the RAMP monitoring package likely contributes to higher signal-to-noise ratios compared to other similar systems, based on discussions with the device manufacturer. However, determination of whether this is the case is beyond the

30 scope of the current paper. The statement has been qualified (16 11-15):

> "The fact that for most gases a variety of calibration approaches show similar (and for typical uses cases, acceptable) performance may reflect better underlying performance from the RAMP monitor, as similar studies for other lowcost sensor packages showed a wider variability in performance between calibration approaches (see e.g. the summary provided by Zimmerman et al., 2018). This suggests that the primary difference between these monitors, i.e. the

internal circuitry which is unique to the RAMP, is the cause for this consistency; however, determination of this is beyond the scope of this paper."

- Citations: twice (Hagan et al., Sadighi et al.) the AMTD citation is used rather than the AMT one.

These have been corrected.

SI: making the data publicly available is a really excellent feature of this paper, and a good template for other sensor papers.
 However the file is almost 14GB! It might make more sense to provide just the raw data, and the scripts used; most users will want the data only. Those that want to examine the model output can run the scripts themselves.

We apologize for the size of the file, however, we felt it was important to include the models themselves, since the randomized nature of the training approach for some models (such as the random forest models) will lead to slightly different results if

10 these models are re-built, as well as a major investment in computational time necessary to re-build all varieties of models considered. However, we have also provided a second version of the data, including only the raw data and scripts but without the calibrated models, which is of a smaller size (about 300MB). Both data sources are referenced in the "Data Availability" section.

**Development of a General Calibration Model and Long-Term Performance Evaluation of Low-Cost Sensors for Air Pollutant Gas Monitoring**

Carl Malings1, Rebecca Tanzer1,3, Aliaksei Hauryliuk1, Sriniwasa P.N. Kumar1, Naomi Zimmerman2, Levent B. Kara3, Albert A. Presto1,3, and R. Subramanian1

[revised manuscript text omitted]

25 following form:

 $c_{\rm CO} = \alpha_{\rm CO} s_{\rm CO} + \alpha_{\rm CO^2} s_{\rm CO}^2 + \alpha_{\rm T} T + \alpha_{\rm T^2} T^2 + \alpha_{\rm RH} RH + \alpha_{\rm RH^2} RH^2 + \alpha_{\rm CO,T} s_{\rm CO} T + \alpha_{\rm CO,RH} s_{\rm CO} RH + \alpha_{\rm T,RH} TRH + \beta_{CO}, (2)$

Note that, as above, a reduced set of inputs is used here. Quadratic regression models using such reduced sets (the same sets used for linear regression) are hereafter referred to as "limited" quadratic regression models; in contrast, models making full use of all available gas, temperature, and humidity sensor inputs from a given RAMP are referred to as "complete" quadratic

30 regression models.

The main advantages of linear and quadratic regression models are their ease of implementation and calibration, as well as their ability to be readily interpreted, e.g., the relative magnitudes of the regression coefficients correspond to the relative

importance of the different inputs in producing the output. The main disadvantage of these models is their inability to compute complicated relationships between input and output which are beyond that of a second-order polynomial. The training and application of linear and quadratic regression models are implemented using custom-written routines for the MATLAB programming language (version R2016b).

**5 2.3.2 Gaussian Process Models**

Gaussian processes are a form of regression which generalizes the multivariate Gaussian distribution to infinite dimensionality (Rasmussen and Williams, 2006). For the purposes of calibration, we make use of a simplified variant of a Gaussian process model. From the training data, both the signals of the RAMP monitors and the readings of the regulatory-grade instruments are transformed such that their distributions during the training period can be approximately modelled as standard normal

10 distributions. This transformation is accomplished by means of a piecewise linear transformation, where the domain is segmented and for each segment different linear mappings are applied. After this transformation, an empirical mean vector  $\mu$ and covariance matrix  $\Sigma$  is computed for the regulatory-grade and RAMP measurements. The transformed measurements can then be described using a multivariate Gaussian distribution. For example, for a RAMP measuring CO, SO2, NO2, O3, and CO2, this distribution would be:

15
$$\left\{c_{C0}', c_{S0_2}', c_{N0_2}', c_{O3}', c_{C0_2}', s_{C0}', s_{S0_2}', s_{N0_2}', s_{O3}', s_{C0_2}', T', RH'\right\}^{\mathrm{T}} \sim \mathcal{N}(\boldsymbol{\mu}, \boldsymbol{\Sigma}),$$
 (3)

where, for example,  $c'_{CO}$  represents the concentration measurement for CO following the transformation. The mean vector and covariance matrix are divided as follows:

$$\boldsymbol{\mu} = \begin{bmatrix} \boldsymbol{\mu}_{conc} \\ \boldsymbol{\mu}_{RAMP} \end{bmatrix} \quad \boldsymbol{\Sigma} = \begin{bmatrix} \boldsymbol{\Sigma}_{conc,conc} & \boldsymbol{\Sigma}_{conc,RAMP} \\ \boldsymbol{\Sigma}_{conc,RAMP}^{T} & \boldsymbol{\Sigma}_{RAMP,RAMP} \end{bmatrix}, \tag{4}$$

where  $\mu_{conc}$  represents the mean of the (transformed) concentration measurements of the regulatory-grade instrument,  $\mu_{RAMP}$ 20 represents the mean of the (transformed) signal measurements from the RAMP,  $\Sigma_{conc,conc}$  represents the covariance of the (transformed) concentrations,  $\Sigma_{RAMP,RAMP}$  represents and covariance of the (transformed) RAMP signals, and  $\Sigma_{conc,RAMP}$ represents the covariance between the (transformed) concentrations and RAMP signals ( $\Sigma_{conc,RAMP}^{T}$  is the transpose of  $\Sigma_{conc,RAMP}$ ). Once these vectors and matrices have been defined, the model is calibrated.

Given a new set of signal measurements from a RAMP, denoted as  $\mathbf{y}_{RAMP} = \{s_{CO}, s_{SO_2}, s_{NO_2}, s_{O_3}, s_{CO_2}, T, RH\}^T$ , these are 25 transformed using the piecewise linear transformation defined above to give the set of transformed signal measures  $\mathbf{y}'_{RAMP}$ . 26 These are then used to estimate the concentrations measured by the RAMP with the standard conditional updating formula of 27 the multivariate Gaussian as follows:

$$c_{\rm CO}', c_{\rm SO_2}', c_{\rm NO_2}', c_{\rm O_3}', c_{\rm CO_2}'\}^{\rm T} = \mu_{\rm conc} + \Sigma_{\rm conc,RAMP} \Sigma_{\rm RAMP,RAMP}^{-1} (\mathbf{y}_{\rm RAMP}' - \mu_{\rm RAMP}),$$
(5)

[revised manuscript text omitted]

---

## Author Comment (AC2) · 3 Dec 2018

We would like to thank the reviewer for their constructive comments. We have tried to address these comments in the attached response document and in the manuscript. Reviewer comments are reproduced in black, our responses are in blue.

**General comments**

In this study, authors compared different models used to adjust data from low-cost sensors and compared model application in different sites within the Pittsburgh, Pennsylvania region of the USA. These comparisons demonstrate that at the sites different model-based calibration techniques show differing abilities to accurately produce concentration data for the various sensors employed. However, there existing two main problems: 1 for calibration period and deployment, it is unclear which based on the text, the calibration data also used for testing or vice versa, more details about the deployment should be illustrated.

All presented results correspond to "testing" data, which is separate from the "training" data used to calibrate the models. This has been clarified in the text (Page 10, Lines 7-9):

> "It should be noted that the metrics presented here are applied only for testing data, i.e., data which were not used to build the calibration models. Model performance on the training data is expected to be higher, and thus less representative of the true capability of the model."

Also, several figures have been added to the supplemental materials illustrating which periods of time are set aside as training and testing data for different sensors. Various other changes have been made throughout the manuscript to clarify what data are being considered, and what models are being applied.

2 Evaluation validity: As listed in Table 4, the average pollutant concentration is relatively small for NO and NO2 both in training and testing period, which about 1.7 and 6.4 ppb level, how did the investigators ensure sensor evaluation validity in such low-level situation, while EC sensor has a lower detection limit at about this level? And for NO, the MAE of all models are even large than average concentration.

Sensor performance was evaluated using only ambient concentrations of pollutants, as their performance in this regime is the most relevant to their performance during actual deployments. This is reflected in the presented results; for example, CvMAE for NO sensors is on the order of 1, indicating absolute errors are of the same magnitude as the readings themselves. We believe this to be an honest way to assess the RAMP sensor and calibration model performance in line with the intended use case of the monitor.

In the specific cases of NO and $NO_2$, we also present the performance of the calibration models at the "Parkway East" site in Figure 4, which was chosen specifically because it typically experiences higher concentrations of these pollutants than our training (CMU) site. In both cases, the performance of the generalized calibration models at these sites (filled markers) was similar to their performance at the training site (hollow markers); thus, we believe the presented performance results to be robust across most typical ambient concentrations which we would encounter in the city of Pittsburgh or any other city with similar climate and air quality characteristics.

In general, we have focused on assessing the relative calibration model performance in terms of CvMAE (e.g. as presented in Figure 2), since we believe this to be readily understandable and to allow easy performance comparisons between sensors measuring different pollutants. However, we have also included information on absolute performance in terms of MAE (e.g.

as presented in Table 5) for interested readers, to allow them to estimate what the expected performance of a similar low-cost sensor package might be in the specific environment they are interested in.

**Specific comments**

1. Page 1, Line 24: Authors use "These results will help guide future efforts in the calibration and use of low-cost sensor systems worldwide". Perhaps the approaches merit many further trials in the widely differing pollutant and meteorological conditions It is unclear. It is unclear how the study demonstrates protocols that apply worldwide?

We believe that the primary conclusion of the paper (i.e. that generalized models for low-cost gas sensor calibration provide comparable performance to individualized models) is widely applicable to any low-cost sensor package with a similar design to the RAMP monitor (including similar electrochemical sensors and appropriate signal conditioning), regardless of where it is used so long as the generalized calibration model is developed for a network of sensors in the region where the network is deployed. We also believe that the methods used to calibrate models for low-cost gas sensor data presented in the paper are widely applicable to any similar monitoring system that produces clean, reproducible raw data of quality comparable to the RAMP output. Although specific performance metrics will likely differ for different locations, meteorological conditions, and low-cost sensor packages, and thus require future testing to assess he performance of specific monitoring packages in specific environments, we do believe that our qualitative results provide important guidance to future low-cost sensor calibration efforts, and that this guidance is generally applicable worldwide.

2. Page 2, line 16: "These sensors tend to have lower signal-to-noise ratios than regulatory-grade instruments" is a fairly unusual and non-specific way to say they are not as precise or sensitive as conventional air monitors.

This has been corrected (Page 2, Lines 17-18):

  "These sensors are less precise and sensitive than regulatory-grade instruments"

3. Page 2, Line 20: Authors mentioned that a nonlinear interaction in the reference (Jiao et al., 2016), however, this paper didn't mention this kind of nonlinear interaction, but cited Gao's paper (Gao et al., 2015) which also didn't mention this, rather it is a nonlinear response for PM sensor.

This citation was incorrect. This has been corrected.

4. Page 2, Line 25: Same problem for reference (Cross et al., 2017) as comment 3.

That paper makes use of high-dimensional model representations involving non-linear component functions (cubic functions are specifically mentioned). We believe this reference is correct.

5. Page 3, line 29: It appears that on-campus field calibrations were conducted during a brief period in summer and early fall. Where winter time calibrations performed as well? If not, how might this impact the application of various results of modelling over winter time conditions?

Field calibrations for some sensors (and for the gRAMP models) included data collected during October, during which conditions were similar to what might be expected during winter (e.g. near-freezing temperatures). Furthermore, time-resolved performance for several sensors across multiple seasons is shown in Figure 7; there is no apparent difference in performance across seasons.

6. At the end Section 2. which describes monitoring methods there is a general reference to the Zimmerman 2018 paper to provide the reader information on protocols. Key factors should be pulled into the current text or figures. Specifically, on Line 30 what were the specific models of regulatory monitors used and how were they located/protected from environmental factors? What methods were employed to measure speciated VOCs mentioned? What is meant by the term "or BTEX"? This does not appear to be covered in the 2018 Zimmerman paper and it does not seem that VOCs are part of this study. Perhaps this needs to be removed from the current text. Further, SO2 is included in the list of pollutants that appear to have been included in this study. However, no data for SO2 appears. Further, several of the models include factors for SO2. Was SO2 measured or employ in this study?

Raw signals from $SO_2$ and VOC sensors within the RAMP monitors are used as potential inputs into the calibration models developed to account for possible cross-sensitivities. However, no calibration models were developed for $SO_2$ or for any VOC species as a part of the work presented in this paper. References to the monitoring of $SO_2$ and VOCs have therefore been removed from this section. For the other gases, the instrument models have been given, and a general description of the location of the instruments has been provided (Page 4, Lines 3-7):

"Less than 10 meters from the RAMP monitors, a suite of high-quality regulatory-grade instruments, measuring ambient concentrations of CO (with a Teledyne T300U instrument), CO2 (LICOR 820), O3 (Teledyne T400 Photometric Ozone Analyser), and NO and NO2 (2B Technologies Model 405nm) are stationed to provide true concentration values for these various gases to which the RAMP monitors are exposed. These regulatory-grade instruments are contained within a mobile laboratory van, into which samples are drawn through an inlet 2.5 meters above ground level."

7. Page 4, Line 5: The collocation in ACHD points seems to span the whole year of 2017 in each week at Lawrenceville as shows in Figure 7, while in Table 4 the maximum days of testing period duration only ranged from 75 to 110 days for CO, O3 and NO2. The authors should give more explanation about the deployment? It is unclear and could not be found in the cited reference (Zimmerman et al., 2018). Also, as mentioned in Page 5, Line 9, 80% of collocation is about 28 days less than 1 month, what are other days while RAMP collocation such long time and what is the criteria for period selection?

Table 4 presents information related to the CMU site only. This has been indicated in the caption. Additional details about how data are divided into training and testing periods are provided in the text of Section 2.3, and figures illustrating these periods have been included in the supplemental information as well.

8. Page 4, Line 10: Authors mentioned ACHD Parkway East has high levels of NO and NO2 for maximum at 100ppb and 40ppb, this period also included in testing period, but in Table 4, for whole testing period these two maximum concentrations are 66 and 32.

Table 4 presents information related to the CMU site only. This has been indicated in the caption.

9. Page 4, Line 22: The best model was selected based on performance of individual models, what is the criteria for this?

Pearson r is used to select the best-performing individual model. This has been clarified in the text (Page 4, Line 33 to Page 5, Line 2):

"Second, from these individualized models, a best individual calibration model (bRAMP) was chosen, which performed best out of all the individualized models on a testing data set with respect to correlation (Pearson r, see Section 2.4)."

10. Page 9, Line 4: The random forest and hybrid model used all sensor data for training CO, and combined linear model which only used CO, why not introduce other sensors in this procedure?

For the linear models, based on prior work by Zimmerman et al., we used only the signal for the target gas of interest (or, in the case of $O_3$, signals from both $NO_2$ and $O_3$ sensors) along with T and RH since only responses to these factors could be reliably modeled as linear. For the quadratic models, we sought to investigate the impacts of including inputs from the other gas sensors in a non-linear model; for this reason, we investigate both "limited" quadratic models, which use the same limited set of inputs as the linear models, and "complete" quadratic models, which use the inputs of all sensors on the RAMP (as do all of the models discussed later, including the hybrid models).

11. Page 10, Line 8: As mentioned in the text "measurements where the corresponding true value is below an assigned lower limit are removed from the measurement set to be evaluated", while for NO2(10ppb) and O3 (10ppb) which listed in Table 1, the training period or testing period concentration range still start from 0 or 1ppb, does any filter about lower limit come into force in the training or evaluation? The authors should discuss the impact of removal of data below specified minimums as this would appear to skew the actual data and do so in a differential basis depending on ambient conditions.

The removal of values below a lower limit is applied only for the evaluation of the Precision and Bias metrics being discussed in this section, and thus only affects the results presented in Figure 5. This has been clarified in the text (Page 11, Lines 7-9):

"Note that this removal of low values is applied only when computing the precision and bias error metrics, and not when evaluating the other metrics described above."

Due to the construction of these metrics, the inclusion of values below these lower bounds would tend to make the Precision and Bias metrics worse. In other words, in ambient conditions where concentrations of pollutants are higher, the Precision and Bias metrics would be better for the same absolute errors. We have chosen to remove the low values as indicated when evaluating these metrics to allow for consistency with other studies which evaluate these metrics in the same way according to EPA recommendations.

12. Page 16, line 11: "The fact that for most gases a variety of calibration approaches show similar and (for typical uses cases, acceptable) performance may reflect better underlying performance from the RAMP monitor, as similar studies for other low-cost sensor packages showed a wider variability in performance between calibration approaches (see e.g. the summary provided by Zimmerman et al., 2018)." This statement appears to indicate that the RAMP monitors somehow performed better than other similar sensor based monitors. However, there is no rationale for this statement based on the actual monitor package, component details provided or information reported by others. It appears to simply be a diffusion based sensor deployment of Alfasense electrochemical cells and an NDIR CO2 sensor. What other factors might influence system performance?

The signal conditioning employed in the RAMP monitoring package likely contribute to higher signal-to-noise ratios compared to other similar systems, based on discussions with the device manufacturer. However, determination of whether this is the case is beyond the scope of the current paper. The statement has been qualified (Page 16, Lines 11-15):

> "The fact that for most gases a variety of calibration approaches show similar (and for typical uses cases, acceptable) performance may reflect better underlying performance from the RAMP monitor, as similar studies for other low-cost sensor packages showed a wider variability in performance between calibration approaches (see e.g. the summary provided by Zimmerman et al., 2018). This suggests that the primary difference between these monitors, i.e. the internal circuitry which is unique to the RAMP, is the cause for this consistency; however, determination of this is beyond the scope of this paper."

13. Page 25, Figure 4: The performance of CMU site for each sensor is marked in hollow marker, while there are two hollow markers in CO and NO2, which is unclear.

The hollow markers are colored to correspond to the RAMPs deployed to each site. For example, the green filled marker represents the performance of a RAMP at the Lawrenceville site, and the hollow green marker indicates the performance of that same RAMP when it was at the CMU site. This was done so that the variability of performance between sites could be compared with the variability in performance between RAMPs at a common site. This has been clarified in the text (Page 13, Lines 3-5):

> "Filled markers indicate the performance of the models at these sites, while hollow markers indicate the 2017 testing period performance of the corresponding RAMP when it was at the CMU site for comparison."

14. Page 30, Table 4: In the evaluation of testing period, R2 was used, however, in previous figure and text, such as Figure 2-4, Pearson linear correlation coefficient r was used, what is the standard for selection between these two parameters?

In reviewing the literature, there was no clear preference for the r or $r^2$ metric to represent correlation. When preparing the paper, we had a slight preference for the r metric, as it distinguishes between positive and negative correlations, which we envisioned might occur for some of the worse-performing calibration models and sensors. Therefore, in figures, the r metric is used. In presenting the results in tabular form, the $r^2$ metric is used, in recognition that some readers may be more familiar with and prefer results presented in that way. For the purposes of these results, where $r^2$ is given for the best-fit line between the calibrated and "true" data, the information provided by r and $r^2$ are the same.

15. Page 30, Table 4: The title of the paper is about a general calibration model (gRAMP) and recommended in the conclusion, while it was not compared with iRAMP models in Table 4. This paper focuses on field calibration methods and model application to produce adjusted pollutant concentration data from sensor-based monitors. However, discussions and recommendations are only made regarding field calibration approaches. It would also appear that the data agreement between sensors and regulatory monitoring might be improved by other means. For example, conventional air monitoring methods as applied to regulatory monitoring efforts always include periodic zero and span challenges. Improved circuitry or air conditioning might also improve monitor performance. How might the inclusion of such methods improve the quality of data produced by sensor-based systems?

Comparisons between iRAMP and gRAMP models are provided in Figures 3 through 6. Data corresponding to the results of Table 4 (now Table 5) for the bRAMP and gRAMP models evaluated at the CMU site have been provided as part of the supplementary materials. The main conclusion of the paper is also based on the relative performance of the gRAMP models when tested at a "new" deployment site (i.e. a site distinct from where the models are trained), as depicted in Figures 4 and 5.

5   The possibility of conducting periodic zero- and span-checks on these sensors was considered but was eventually dismissed due to the logistical challenges of performing these checks over a large network of low-cost sensors deployed over a relatively large spatial domain. Improved circuitry within the RAMP monitor to improve signal-to-noise ratios might be a possibility but was not investigated as a part of this paper (discussions with the manufacturer indicated that these monitors already have very good signal conditioning electronics compared to other comparable products). Providing climate control for the RAMP monitor

10  would greatly increase the cost and power consumption of the monitor, and so was not considered. Ongoing work is instead being focused on different automated methods of in-field calibration checks, for example by comparing with nearby regulatory monitoring sites, by comparing with data collected from mobile sampling campaigns, and by using clusters of nearby monitors in an attempt to identify and correct for outliers.

[revised manuscript text omitted]
}_{\mathrm{conc,conc}} & \boldsymbol{\Sigma}_{\mathrm{conc,RAMP}} \\ \boldsymbol{\Sigma}^{\mathrm{T}}_{\mathrm{conc,RAMP}} & \boldsymbol{\Sigma}_{\mathrm{RAMP,RAMP}} \end{bmatrix}, \tag{4}$$

where $\boldsymbol{\mu}_{\mathrm{conc}}$ represents the mean of the (transformed) concentration measurements of the regulatory-grade instrument, $\boldsymbol{\mu}_{\mathrm{RAMP}}$ represents the mean of the (transformed) signal measurements from the RAMP, $\boldsymbol{\Sigma}_{\mathrm{conc,conc}}$ represents the covariance of the (transformed) concentrations, $\boldsymbol{\Sigma}_{\mathrm{RAMP,RAMP}}$ represents and covariance of the (transformed) RAMP signals, and $\boldsymbol{\Sigma}_{\mathrm{conc,RAMP}}$ represents the covariance between the (transformed) concentrations and RAMP signals ($\boldsymbol{\Sigma}^{\mathrm{T}}_{\mathrm{conc,RAMP}}$ is the transpose of $\boldsymbol{\Sigma}_{\mathrm{conc,RAMP}}$). Once these vectors and matrices have been defined, the model is calibrated.

Given a new set of signal measurements from a RAMP, denoted as $\mathbf{y}_{\mathrm{RAMP}} = \{s_{CO}, s_{SO_2}, s_{NO_2}, s_{O_3}, s_{CO_2}, T, RH\}^{\mathrm{T}}$, these are transformed using the piecewise linear transformation defined above to give the set of transformed signal measures $\mathbf{y}'_{\mathrm{RAMP}}$. These are then used to estimate the concentrations measured by the RAMP with the standard conditional updating formula of the multivariate Gaussian as follows:

$$\{c'_{CO}, c'_{SO_2}, c'_{NO_2}, c'_{O_3}, c'_{CO_2}\}^{\mathrm{T}} = \boldsymbol{\mu}_{\mathrm{conc}} + \boldsymbol{\Sigma}_{\mathrm{conc,RAMP}} \boldsymbol{\Sigma}^{-1}_{\mathrm{RAMP,RAMP}} (\mathbf{y}'_{\mathrm{RAMP}} - \boldsymbol{\mu}_{\mathrm{
[revised manuscript text omitted]

| Gas | Training Period | | | | Testing Period | | | |
| --- | --- | --- | --- | --- | --- | --- | --- | --- |
| | Duration | Concentration | | | Duration | Concentration | | |
| | [days] | [ppb] | | | [days] | [ppb] | | |
| | | *lower range* | *average range* | *upper range* | | *lower range* | *average range* | *upper range* |
| | *Range* | | | | *Range* | | | |
| CO | 21 - 28 | 57-118 | 193-356 | 923-3750 | 3 - 75 | 57-120 | 145-451 | 235-3750 |
| NO | 26 - 28 | 0-1 | 1-4 | 21-66 | 4 - 93 | 0-2 | 1-3 | 11-66 |
| $NO_2$ | 22 - 28 | 0-1 | 5-9 | 19-31 | 4 - 110 | 0-1 | 4-9 | 15-32 |
| $O_3$ | 21 - 28 | 1-3 | 21-36 | 62-128 | 2 - 76 | 1-23 | 22-48 | 54-128 |
| T | [°C] | 2-16 | 18-26 | 32-42 | [°C] | 0-18 | 14-27 | 27-42 |
| RH | [%] | 26-52 | 56-71 | 66-94 | [%] | 25-52 | 50-73 | 64-94 |

**Table 5: Performance data for iRAMP models at CMU in 2017 (Avg. is the average, SD is the standard deviation). The "#" sub-column under "Model" indicates the total number of iRAMP models developed for each gas. Slope and $r^2$ are presented for the best-fit-line between the calibrated RAMP measures and those of the regulatory monitor.**

| Gas | Model Type | # | Slope | | $r^2$ | | MAE [ppb] | | Bias [ppb] | |
|---|---|---|---|---|---|---|---|---|---|---|
| | | | Avg. | SD | Avg. | SD | Avg. | SD | Avg. | SD |
| CO | LR | 48 | 1.08 | 0.36 | 0.75 | 0.16 | 60 | 19 | -6 | 18 |
| | LQR | 48 | 1.01 | 0.18 | 0.83 | 0.10 | 48 | 11 | -5 | 14 |
| | CQR | 48 | 0.96 | 0.16 | 0.85 | 0.09 | 46 | 16 | -3 | 20 |
| | CL | 48 | 1.06 | 0.24 | 0.74 | 0.11 | 58 | 17 | 1 | 22 |
| | NN | 48 | 1.33 | 1.11 | 0.46 | 0.23 | 84 | 34 | -2 | 34 |
| | HY | 48 | 0.94 | 0.20 | 0.77 | 0.12 | 52 | 15 | 11 | 22 |
| NO | LR | 19 | 1.31 | 0.56 | 0.15 | 0.07 | 2.3 | 1.1 | 0.26 | 0.80 |
| | LQR | 19 | 1.15 | 0.52 | 0.25 | 0.15 | 2.3 | 1.1 | 0.26 | 0.80 |
| | CQR | 19 | 0.97 | 0.36 | 0.36 | 0.14 | 2.1 | 1.0 | 0.35 | 0.82 |
| | CL | 19 | 0.67 | 0.33 | 0.18 | 0.10 | 2.2 | 1.2 | 0.08 | 0.80 |
| | NN | 19 | 0.90 | 0.35 | 0.30 | 0.13 | 2.0 | 1.0 | 0.09 | 0.55 |
| | HY | 19 | 0.65 | 0.37 | 0.32 | 0.14 | 2.3 | 0.8 | 0.76 | 0.66 |
| $NO_2$ | LR | 62 | 0.89 | 0.31 | 0.17 | 0.08 | 3.4 | 0.7 | 0.16 | 0.88 |
| | LQR | 62 | 0.79 | 0.23 | 0.21 | 0.09 | 3.3 | 0.6 | 0.18 | 0.93 |
| | CQR | 62 | 0.85 | 0.15 | 0.47 | 0.11 | 2.6 | 0.5 | 0.07 | 0.73 |
| | CL | 62 | 0.77 | 0.17 | 0.37 | 0.12 | 2.9 | 0.5 | 0.27 | 0.70 |
| | NN | 62 | 0.93 | 0.16 | 0.49 | 0.12 | 2.6 | 0.5 | 0.07 | 0.59 |
| | HY | 62 | 0.83 | 0.13 | 0.48 | 0.10 | 2.6 | 0.4 | 0.51 | 0.63 |
| $O_3$ | LR | 44 | 0.98 | 0.06 | 0.80 | 0.12 | 5.1 | 1.7 | -0.05 | 1.6 |
| | LQR | 44 | 0.96 | 0.05 | 0.83 | 0.11 | 4.6 | 1.7 | 0.12 | 1.4 |
| | CQR | 44 | 0.93 | 0.07 | 0.82 | 0.12 | 4.6 | 1.7 | -0.08 | 1.1 |
| | CL | 44 | 0.89 | 0.11 | 0.62 | 0.11 | 7.3 | 1.3 | -0.47 | 2.4 |
| | NN | 44 | 0.98 | 0.21 | 0.73 | 0.26 | 5.8 | 2.8 | 0.09 | 1.3 |
| | HY | 44 | 0.93 | 0.06 | 0.81 | 0.09 | 4.9 | 1.3 | 0.40 | 1.5 |

[Figure]

**Figure S1. Results corresponding to Fig. 2 for CO₂.**

[Figure]

**Figure S2. Results corresponding to Fig. 3 for CO₂.**

[Figure]

**Figure S3. Results corresponding to Fig. 6 for CO₂.**

[Figure]

**Figure S4. An evaluation of the performance of the calibration algorithms as a function of the averaging period applied to the raw RAMP data. All models are trained using data collected at the CMU site in 2017. Performance is also evaluated at the CMU site in 2017. Solid lines indicate median performance across RAMPs, dashed lines indicate 25th and 75th percentiles of performance. For CO, the gRAMP LQR model is used; for $O_3$ the gRAMP HY model is used; for $CO_2$ the gRAMP HY model is used. Note that all models were originally developed using data averaged at 15 minutes. Results are presented for CvMAE and Pearson r, for averaging times ranging from 1 minute up to 1 hour. These results indicate that the performance of the calibration approaches are fairly stable**

**for data averaged over periods ranging up to 1 hour. At longer averaging periods, the use of time-averaged environmental variables (such as temperature and relative humidity) in the calibration model appears to reduce performance.**

[Figure]

5    **Figure S5. An evaluation of the performance of the calibration algorithms as a function of the averaging period; in contrast to the previous figure, this presents the results when averaging is performed after calibration, rather than before. In terms of CvMAE, performance improves as averaging time increases. In terms of Pearson r, results can be worse with longer averaging, due to the**

reduction in the number of points used to evaluate correlation (since there are fewer time periods overall to compare) and to the reduction in the variability (although accuracy is improving as averaging time increases, the variability in the data are also being reduced, and so correlation is decreasing).

[Figure]

5  **Figure S6. Description of the training and testing periods used for CO models. Blue bars indicate periods used for training data, while red bars denote periods set aside for testing. Time divisions for individual RAMPs (with numeric IDs) are presented corresponding to data used for iRAMP and bRAMP models. Divisions for the "gen" RAMP indicate the training data periods used for gRAMP models, derived from the median of data from the training set of RAMPs collected during these periods; testing data for gRAMP models is drawn from RAMPs which are not part of the training set of RAMPs.**

[Figure]

Figure S7. Description of the training and testing periods used for NO models. Blue bars indicate periods used for training data, while red bars denote periods set aside for testing. Time divisions for individual RAMPs (with numeric IDs) are presented corresponding to data used for iRAMP and bRAMP models. Divisions for the "gen" RAMP indicate the training data periods used for gRAMP models, derived from the median of data from the training set of RAMPs collected during these periods; testing data for gRAMP models is drawn from RAMPs which are not part of the training set of RAMPs.

[Figure]

**Figure S8. Description of the training and testing periods used for NO₂ models.** Blue bars indicate periods used for training data, while red bars denote periods set aside for testing. Time divisions for individual RAMPs (with numeric IDs) are presented corresponding to data used for iRAMP and bRAMP models. Divisions for the "gen" RAMP indicate the training data periods used for gRAMP models, derived from the median of data from the training set of RAMPs collected during these periods; testing data for gRAMP models is drawn from RAMPs which are not part of the training set of RAMPs.

[Figure]

**Figure S9. Description of the training and testing periods used for O₃ models. Blue bars indicate periods used for training data, while red bars denote periods set aside for testing. Time divisions for individual RAMPs (with numeric IDs) are presented corresponding to data used for iRAMP and bRAMP models. Divisions for the "gen" RAMP indicate the training data periods used for gRAMP models, derived from the median of data from the training set of RAMPs collected during these periods; testing data for gRAMP models is drawn from RAMPs which are not part of the training set of RAMPs.**

[Figure]

**Figure S10. Description of the training and testing periods used for CO₂ models.** Blue bars indicate periods used for training data, while red bars denote periods set aside for testing. Time divisions for individual RAMPs (with numeric IDs) are presented corresponding to data used for iRAMP and bRAMP models. Divisions for the "gen" RAMP indicate the training data periods used for gRAMP models, derived from the median of data from the training set of RAMPs collected during these periods; testing data for gRAMP models is drawn from RAMPs which are not part of the training set of RAMPs.

[Figure]

**Figure S11. Depiction of the range of CO concentrations experienced during training and testing. Blue boxplots indicate training ranges, while red boxplots denote testing ranges. Dots with circles indicate the midpoint, thicker bars indicate the interquartile range, thinner bars show 1st and 99th percentiles, and colored dots depict outliers. The horizontal axis shows the RAMP ID number (or "gen", which depicts the concentration range used for training gRAMP models).**

[Figure]

**Figure S12. Depiction of the range of NO concentrations experienced during training and testing. Blue boxplots indicate training ranges, while red boxplots denote testing ranges. Dots with circles indicate the midpoint, thicker bars indicate the interquartile range, thinner bars show 1st and 99th percentiles, and colored dots depict outliers. The horizontal axis shows the RAMP ID number (or "gen", which depicts the concentration range used for training gRAMP models).**

[Figure]

**Figure S13. Depiction of the range of NO₂ concentrations experienced during training and testing. Blue boxplots indicate training ranges, while red boxplots denote testing ranges. Dots with circles indicate the midpoint, thicker bars indicate the interquartile range, thinner bars show 1st and 99th percentiles, and colored dots depict outliers. The horizontal axis shows the RAMP ID number (or "gen", which depicts the concentration range used for training gRAMP models).**

[Figure]

**Figure S14. Depiction of the range of O₃ concentrations experienced during training and testing. Blue boxplots indicate training ranges, while red boxplots denote testing ranges. Dots with circles indicate the midpoint, thicker bars indicate the interquartile range, thinner bars show 1st and 99th percentiles, and colored dots depict outliers. The horizontal axis shows the RAMP ID number (or "gen", which depicts the concentration range used for training gRAMP models).**

[Figure]

**Figure S15. Depiction of the range of CO₂ concentrations experienced during training and testing. Blue boxplots indicate training ranges, while red boxplots denote testing ranges. Dots with circles indicate the midpoint, thicker bars indicate the interquartile range, thinner bars show 1st and 99th percentiles, and colored dots depict outliers. The horizontal axis shows the RAMP ID number (or "gen", which depicts the concentration range used for training gRAMP models).**

**Table S1: Results corresponding to Table 4 for CO₂.**

| Gas | Training Period | | | | Testing Period | | | |
|---|---|---|---|---|---|---|---|---|
| | Duration | Concentration | | | Duration | Concentration | | |
| | [days] | [ppm] | | | [days] | [ppm] | | |
| | | *lower* | *average* | *upper* | | *lower* | *average* | *upper* |
| | *Range* | *range* | *range* | *range* | *Range* | *range* | *range* | *range* |
| CO₂ | 21 - 28 | 365-384 | 413-454 | 528-567 | 2 - 50 | 365-388 | 399-458 | 471-601 |

**Table S2: Results corresponding to Table 5 for CO₂.**

| Gas | Model Type | # | Testing Performance | | | | | | | |
|---|---|---|---|---|---|---|---|---|---|---|
| | | | Slope | | r² | | MAE [ppm] | | Bias [ppm] | |
| | | | *Avg.* | *SD* | *Avg.* | *SD* | *Avg.* | *SD* | *Avg.* | *SD* |
| CO₂ | LR | 38 | 0.74 | 0.23 | 0.21 | 0.09 | 24 | 5 | 0.6 | 6.1 |
| | LQR | 38 | 0.62 | 0.22 | 0.23 | 0.12 | 25 | 5 | 1.0 | 8.1 |
| | CQR | 38 | 0.74 | 0.20 | 0.47 | 0.16 | 18 | 3 | -1.0 | 4.8 |
| | CL | 38 | 0.76 | 0.13 | 0.43 | 0.13 | 20 | 3 | 1.4 | 4.4 |
| | NN | 38 | 0.47 | 1.53 | 0.28 | 0.25 | 23 | 7 | -2.1 | 6.5 |
| | HY | 38 | 0.79 | 0.26 | 0.53 | 0.15 | 19 | 4 | 3.2 | 6.0 |

**Table S3: Results corresponding to Table 5 for bRAMP models.**

| Gas | Model Type | # | Slope Avg. | Slope SD | r² Avg. | r² SD | MAE [ppb] Avg. | MAE [ppb] SD | Bias [ppb] Avg. | Bias [ppb] SD |
|-----|-----|---|------|------|------|------|------|------|------|------|
| CO | LR | 1 | 0.68 | 0.28 | 0.66 | 0.23 | 80 | 38 | 54 | 138 |
|  | LQR | 1 | 0.75 | 0.25 | 0.71 | 0.21 | 59 | 28 | 56 | 119 |
|  | CQR | 4 | 0.62 | 0.29 | 0.58 | 0.29 | 143 | 337 | 156 | 568 |
|  | CL | 4 | 0.83 | 0.31 | 0.64 | 0.23 | 66 | 29 | 63 | 78 |
|  | NN | 4 | 1.09 | 0.58 | 0.46 | 0.20 | 81 | 31 | 42 | 90 |
|  | HY | 4 | 0.74 | 0.26 | 0.69 | 0.24 | 63 | 38 | 98 | 90 |
| NO | LR | 1 | 0.60 | 0.67 | 0.09 | 0.09 | 8.3 | 20.6 | 8.9 | 19.6 |
|  | LQR | 1 | 0.69 | 0.53 | 0.14 | 0.14 | 3.8 | 7.2 | 1.9 | 6.5 |
|  | CQR | 2 | 0.55 | 0.44 | 0.19 | 0.17 | 8.0 | 23.2 | 5.4 | 19.2 |
|  | CL | 2 | 0.38 | 0.35 | 0.09 | 0.13 | 3.1 | 1.7 | 1.1 | 1.4 |
|  | NN | 2 | 1.00 | 0.57 | 0.21 | 0.16 | 2.2 | 1.1 | 0.2 | 2.0 |
|  | HY | 2 | 0.58 | 0.31 | 0.26 | 0.15 | 2.5 | 1.8 | 1.4 | 2.2 |
| NO₂ | LR | 1 | 0.75 | 0.42 | 0.14 | 0.10 | 4.7 | 9.6 | -0.2 | 7.0 |
|  | LQR | 1 | 0.64 | 0.27 | 0.18 | 0.12 | 3.5 | 0.9 | -1.4 | 1.8 |
|  | CQR | 4 | 0.44 | 0.35 | 0.25 | 0.19 | 4.2 | 2.1 | 2.1 | 4.4 |
|  | CL | 4 | 0.58 | 0.29 | 0.21 | 0.14 | 3.5 | 0.6 | 2.1 | 2.9 |
|  | NN | 4 | 0.86 | 0.37 | 0.33 | 0.18 | 3.1 | 0.7 | 0.5 | 2.4 |
|  | HY | 4 | 0.78 | 0.30 | 0.32 | 0.19 | 3.2 | 1.2 | 1.4 | 2.5 |
| O₃ | LR | 1 | 0.76 | 0.24 | 0.70 | 0.23 | 10.5 | 24.8 | 4.7 | 21.5 |
|  | LQR | 1 | 0.85 | 0.22 | 0.72 | 0.24 | 6.1 | 3.1 | -3.1 | 8.9 |
|  | CQR | 2 | 0.75 | 0.27 | 0.65 | 0.27 | 9.8 | 17.1 | 2.9 | 15.5 |
|  | CL | 2 | 0.91 | 0.30 | 0.50 | 0.18 | 8.7 | 2.1 | -1.7 | 8.2 |
|  | NN | 2 | 0.90 | 0.53 | 0.65 | 0.26 | 7.0 | 3.9 | -2.6 | 7.4 |
|  | HY | 2 | 1.06 | 0.21 | 0.75 | 0.13 | 5.8 | 1.8 | 0.3 | 6.5 |
| CO₂ | LR | 1 | 0.65 | 0.40 | 0.18 | 0.14 | 23 | 5 | 13 | 19 |
|  | LQR | 1 | 0.41 | 0.32 | 0.16 | 0.16 | 27 | 9 | 15 | 25 |
|  | CQR | 4 | 0.43 | 0.27 | 0.31 | 0.21 | 58 | 171 | 17 | 148 |
|  | CL | 4 | 0.70 | 0.18 | 0.32 | 0.15 | 21 | 4 | 9 | 17 |
|  | NN | 4 | 0.79 | 0.53 | 0.29 | 0.18 | 31 | 32 | -9 | 46 |
|  | HY | 4 | 0.95 | 0.27 | 0.47 | 0.16 | 18 | 3 | 12 | 17 |

**Table S4: Results corresponding to Table 5 for gRAMP models.**

| Gas | Model Type | # | Slope Avg. | Slope SD | r² Avg. | r² SD | MAE [ppb] Avg. | MAE [ppb] SD | Bias [ppb] Avg. | Bias [ppb] SD |
|-----|------------|---|------|------|------|------|------|------|------|------|
| CO | LR | 1 | 1.03 | 0.24 | 0.80 | 0.11 | 68 | 12 | 26 | 109 |
| | LQR | 1 | 0.90 | 0.08 | 0.85 | 0.09 | 56 | 8 | 6 | 93 |
| | CQR | 1 | 0.69 | 0.18 | 0.66 | 0.20 | 106 | 41 | 60 | 95 |
| | CL | 1 | 1.02 | 0.19 | 0.72 | 0.10 | 80 | 12 | 21 | 57 |
| | NN | 1 | 0.67 | 0.22 | 0.51 | 0.17 | 134 | 59 | 88 | 111 |
| | HY | 1 | 0.75 | 0.11 | 0.61 | 0.11 | 110 | 41 | 75 | 54 |
| NO | LR | 1 | 1.51 | 0.92 | 0.07 | 0.03 | 3.8 | 1.8 | 0.1 | 1.8 |
| | LQR | 1 | 0.67 | 0.34 | 0.08 | 0.04 | 7.1 | 9.6 | 3.5 | 8.5 |
| | CQR | 1 | 0.15 | 0.09 | 0.06 | 0.04 | 5.9 | 6.4 | 2.7 | 6.3 |
| | CL | 1 | 0.43 | 0.13 | 0.13 | 0.03 | 3.1 | 1.0 | -0.6 | 0.5 |
| | NN | 1 | 0.49 | 0.23 | 0.22 | 0.12 | 4.1 | 3.0 | 2.3 | 4.9 |
| | HY | 1 | 0.40 | 0.22 | 0.17 | 0.08 | 13.2 | 22.7 | 9.9 | 20.1 |
| $NO_2$ | LR | 1 | 1.07 | 0.40 | 0.14 | 0.05 | 3.9 | 0.7 | -1.2 | 1.5 |
| | LQR | 1 | 0.86 | 0.31 | 0.18 | 0.07 | 3.8 | 0.7 | -1.1 | 1.9 |
| | CQR | 1 | 0.67 | 0.21 | 0.30 | 0.09 | 3.5 | 0.4 | -1.0 | 2.7 |
| | CL | 1 | 0.67 | 0.19 | 0.26 | 0.12 | 3.6 | 0.5 | -0.1 | 2.4 |
| | NN | 1 | 0.88 | 0.21 | 0.34 | 0.15 | 3.3 | 0.4 | -0.3 | 2.9 |
| | HY | 1 | 0.76 | 0.27 | 0.30 | 0.17 | 3.4 | 0.5 | 0.2 | 2.6 |
| $O_3$ | LR | 1 | 0.89 | 0.27 | 0.72 | 0.22 | 6.4 | 2.8 | 2.2 | 4.9 |
| | LQR | 1 | 0.77 | 0.29 | 0.66 | 0.24 | 7.5 | 4.1 | 4.4 | 6.8 |
| | CQR | 1 | 0.76 | 0.28 | 0.67 | 0.25 | 7.2 | 4.2 | 3.5 | 6.5 |
| | CL | 1 | 0.91 | 0.13 | 0.45 | 0.19 | 8.9 | 1.8 | -0.8 | 5.7 |
| | NN | 1 | 0.92 | 0.18 | 0.73 | 0.18 | 5.9 | 2.4 | 1.7 | 5.1 |
| | HY | 1 | 1.00 | 0.12 | 0.73 | 0.12 | 5.9 | 1.5 | 0.9 | 2.9 |
| $CO_2$ | LR | 1 | 0.63 | 0.14 | 0.21 | 0.06 | 26 | 4 | -2 | 12 |
| | LQR | 1 | 0.55 | 0.14 | 0.20 | 0.06 | 27 | 4 | -6 | 12 |
| | CQR | 1 | 0.39 | 0.11 | 0.22 | 0.11 | 27 | 5 | -9 | 15 |
| | CL | 1 | 0.71 | 0.15 | 0.37 | 0.13 | 23 | 2 | 3 | 16 |
| | NN | 1 | 0.30 | 0.25 | 0.15 | 0.12 | 38 | 25 | -12 | 19 |
| | HY | 1 | 0.80 | 0.16 | 0.43 | 0.11 | 21 | 2 | 4 | 15 |

---

## Referee Report (RR1)

**Review Comments on the Revised Manuscript**

**General comments**

The authors have responded well to most of the comments from the reviewers. And the paper is much improved.

A few important issue remain that were not fully addressed in responses. These are covered in detail in the specific comments below. However, generally, there remains an issue with cold weather operations data/fittings and with comments/conclusions regarding the application of findings to other sensors/situations. This work was performed in a specific location in the US and essential data were collected during a period of summer and early fall. A specific brand of sensor cells was employed. These findings don't justify sweeping conclusions for application in other situations and with other sensor configurations. A further quick review is recommended following consideration of the points raised since the answers may impact the findings.

Page 3, line 4—the description includes the word "affordable" but nowhere in the text is the cost of the RAMP system (with all sensors) mentioned. These appear to be available for purchase from a company called "SenSevere".

Line 4, line 8--I am not convinced that the role of temp/RH are well established in the study. While the authors responded to the earlier question with a statement that they ran "into October" and it was suggested to have similar conditions as winter a quick look at weather underground for the monthly average temp for October shows 54 degrees while January is 26. The humidity is also likely to vary far outside the range encountered during summer/early fall months. Our experience with Alphasense based sensors is that these differences are important. Basically it appears that a great deal of good data are reported and used from the limited period, but the cold months (which may have differing RH) are untested. The authors need to explicitly discuss this and what they may know about cold period operations. The exact date of the range of operations should also be presented.

Responses to earlier questions directs one to see figure 7 to demonstrate agreement during cold seasons, but this appears to show that performance is not equivalent between winter and warm season observations for NO2. This difference is not discussed in the paper. For Ozone there does not appear to be very much actual cold season operation and one would also expect quite low ozone in any case, to help judge model fitting.

Are these platforms heated or is internal temperature measured/reported? What temp/RH data are used in fittings?

Does the present data set and modelling support the overall summary statements regarding the suitability of the model results/application advice in the middle of page 9 which did not include cold season data?

Page 5, line 32—There are several monitoring platforms based on "low cost sensor" being used/sold. While many use Alphasense sensors, others are used as well. A bit more care is needed with the general statement regarding application of methods to unspecific low cost electrochemical sensors. They may or may not respond the same.

Page 20, line one—the observation/finding that specific fitting models work well "when applied to data collected at new sites" seems a bit overstated based on the very limited geographical/meteorological diversity studied here.

Line 10—the statement which proposes the reason for good data quality is proposed as "This suggests that the primary difference between these monitors, i.e. the internal circuitry which is unique to the RAMP, is the cause for this consistency" but there seems to be no description of the circuitry to justify this statement. Is it based on Alphasense reference circuity or some other improved design. Please expand and support this statement or remove it. Perhaps the circuitry might be described in the methods section.

Page 21—line 4. All work reported is from Alphasense B4 sensors. Is it clear that the finding and recommendations made in this study apply to all electrochemical cells?

Page 22, line 15. Conclusions. It seems unwise, based on the reports from this study that sensor responses may drop by half over a year, to only calibrate on an annual basis. This is a lot of sensor "drift"! It is unclear that the rate of decay is linear and that methods exist to determine calibration factors over the course of the year. Further discussion and data are needed on this.

---

## Author Response (AR2)

We would like to again thank the editor and reviewers for their constructive comments. We have tried to address these comments in the attached response document and in the manuscript. Comments are reproduced in black, our responses are in blue.

**Associate Editor**

Comments to the Author:

One review recommends 'publish as it is', and another recommends 'minor changes' as suggested in below. Please address the recommendations before finalizing the paper for submission and publication.

peer reviews:

This is an improved manuscript – in particular, the additional details about the training and test data sets have substantially clarified the authors' approaches to sensor calibration. Most of my concerns have been addressed; remaining comments are listed below.

If training and test sets are not the same (p. 6, line 1), it's hard to understand the utility (or even the meaning) of Table 4 – combining all test and training data into combined ranges isn't terribly meaningful, since there's substantial overlap between the two. Moreover combining them even risks misleading the reader as to the true ranges of the two sets for individual sensors – the ranges for an individual sensor might be quite different than those shown in the table. I would recommend changing this table substantially to make clear the differences in training/test sets for individual sensors. In addition, these differences should be discussed in the table caption. If there was a way to visualize these differences (maybe some example histograms?) that would be helpful also.

Table 4 lists ranges of upper, lower, and average concentration values across all sensors. This gives an indication of the variability in these concentrations across the RAMPs. For example, for $O_3$, average concentrations during the training period can vary from 21 to 36 ppb, while the maximum 15-minute-average concentration for the training period can vary between 62 and 128 ppb across RAMPs. Supplemental information figures S11 through S15 provide boxplots depicting the ranges of values for the training and testing set used for each gas and sensor to provide more specific information. The caption for the table has been expanded to better explain this, and make reference to the supplemental information for more details:

> "Table 4: Durations and ranges of testing and training data at CMU in 2017. Durations of the training and testing periods are in days. Ranges indicated are in ppb for all gases, degrees Celsius for temperature (T), and percent for relative humidity (RH). Note that because training and testing periods vary for different RAMPs, as described in Section 2.3, the duration and concentration ranges of the training and testing periods will likewise vary. This table gives an indication of the variability in training and testing period durations across RAMPs, as well as the variability in concentrations, temperature, and relative humidity. The "lower range" indicates the variability in the lowest 15-minute-average concentration experienced by RAMPs during training or testing. Likewise, the "upper range" indicates the variability in highest 15-minute-average concentrations. The "average range" indicates the variability in the average 15-minute-average concentration across RAMPs for either the training or testing period. Further information about these ranges is provided in the supplementary information (see Figures S11-S15)."

Results: in my original review I had suggested that the random forest and hybrid approaches shouldn't be different, since the training and test sets appeared to be identical. But since the ranges given in Table 4 turn out to be combined ranges, and not ranges covered by each individual sensor, this may be incorrect – there may be sensors for which the ranges of the training and test sets differ substantially. (However, whether this is actually the case is hard to evaluate based on the information in the paper and SI.) In such cases the two models may be expected to give different results, so the two could be discussed individually.

Regardless, if hybrid model is to be left in, the authors should still need to provide information on the number of "crossings" between the RF and LR models, and the fraction time evaluated by RF vs time evaluated by LR.

Although ranges vary from RAMP to RAMP, for a given RAMP the same range will be used for the training of all algorithms, including random forest and hybrid models. However, since the performance of the random forest model on calibration of RAMP data has already been discussed in previous work (Zimmerman et al., 2018), we will continue to refrain from including it in the current manuscript.

Information has been included about the active times of the random forest model as part of the hybrid approach, as well as the average interval between "crossings" from the random forest to linear models (P. 12, lines 7-11):

> "For the hybrid models, across gases, their random forest components were typically active from 88% to 93% of the time (or one "crossing" from random forest to linear models every 12 to 17 hours of active sensor time), although for specific RAMPs this ranged from 75% to 99% (one "crossing" every 5 to 83 hours) depending on the ranges of training and testing data. For perspective, the random forest component would be active 90% of the time if the distributions of training and testing concentrations were identical."

P. 16, lines 9-11: it should also be mentioned that the clustering algorithm would likely be improved by use of kNN (rather than k-means-clustering, which is what is used).

This has been added (P. 16, lines 9-11):

> "These results could perhaps be improved further; for instance, our linear and quadratic regression models did not use regularization, nor did we experiment with neural networks involving multiple hidden layers and varying numbers of nodes, nor with the use of different k-means and nearest-neighbor algorithms for clustering."

P. 16, lines 13-15: if the authors are going to continue to make this suggestion based on the current work (even with the new caveat added), they need to be back it up with much more than a citation to another paper. Specifically they need to show some evidence that the differences in performance results from the RAMP circuitry, and not from differences in the training/test set used. I'm not sure how one would do this, but as written the sentence is purely speculative, and not backed up with any substantive evidence.

We will leave the evaluation of the RAMP circuitry performance as a topic for future work. We have therefore removed this statement.

P. 18 line 14: it is stated that a new model should be developed "each year", but this is probably more specific than is warranted from the work. My takeaway from this work is that models stay reasonably robust for timescales of several months, but should be periodically evaluated/updated when used over longer timescales (on the order of every ~6-18 months). I would recommend changing to wording to reflect this. This recommendation is also included in the abstract,

and so should be changed there as well. (As a minor side note, I feel including it in the abstract risks detracting from the more fundamental results of this work, related to generalized models. So I might recommend removing or shortening the sentence in lines 20-22 of the abstract.)

The recommendation has been updated (P. 18, lines 10-13):

> "For long-term deployments, it is recommended that model performance be periodically re-evaluated (using limited co-location campaigns with a subset of the deployed sensors) every 6 months to one year, and that development of new calibration models be contingent on the outcomes of these re-evaluations. This recommendation is due to the noticeable change in performance when models for one year were used for processing data collected in the subsequent year."

The sentence in the abstract has also been changed (P. 1, lines 20-22):

> "For long-term deployments, it is recommended that model performance be re-evaluated and new models developed periodically, due to the noticeable change in performance over periods of a year or more."

SI: in the Response to Reviews the authors state that "the randomized nature of the training approach for some models (such as the random forest models) will lead to slightly different results if these models are re-built." I don't understand this. If the algorithm uses a random seed to generate psuedo-random numbers, the same psuedo-random numbers should be generated each time, so results should be replicable. (More generally, if the results are indeed different when the model is re-run, this represents a potentially major problem, as it calls into question the robustness of the results.)

Pseudo-random number generation is performed within the MATLAB packages related to neural network and random forest model calibration, and so we cannot guarantee that the exact same results will be achieved every time, due to the pseudo-random number generation being performed within these pre-existing packages. However, the resulting differences should not be enough to noticeably affect the results, and are (based on some tests conducted by the authors) several orders of magnitude lower than the inter-RAMP variability presented for the results.